# Efficient Controlled Language Generation with Low-Rank Autoregressive Reward Models

## Abstract

Language models trained on large amounts of data are known to produce inappropriate content in some cases and require careful tuning to be used in the real world. We revisit the reward augmented decoding (RAD) approach to control the generation from a language model using the scores from a task-specific reward model. We investigate the training objective of RAD, and reformulate it as a task of learning a reward matrix. We show that RAD is designed to support high flexibility when representing the reward matrices, which leads to higher computational costs during decoding. However, we demonstrate that RAD does not use its full flexibility. Motivated by this, we propose a simpler but more efficient low-rank parametrization of the reward model enabling fast and effective guided decoding. For the detoxification and sentiment control tasks, we show that our low-rank reward model performs on par with the more flexible RAD parametrization, while requiring only a single reward model call per generated token.

## 1 Introduction

Generative large language models (LLMs) have gained a lot of popularity in recent years and shown impressive results in zero-shot and few-shot scenarios on numerous downstream tasks (Touvron et al., 2023; OpenAI, 2024; Jiang et al., 2023). These large-scale models are pretrained on large amounts of data, and are known to inherit and memorize underlying biases (Sheng et al., 2019) as well as to provide unsafe responses (Wallace et al., 2019; Ganguli et al., 2022), necessitating further tuning for safer deployment and control (Ouyang et al., 2022).

Control over LLMs can be roughly divided into methods which modify the original model via finetuning (Ouyang et al., 2022; Rafailov et al., 2023), and decoding-time solutions, which do not modify the parameters of the original model. As models increase in size, finetuning becomes prohibitive with limited computational resources. In this work, we focus on a more modular approach of decoding-time guidance, and assume we have access to top-$k$ logits of a black-box base language model (see §2.1 for details). In this line of work, a discriminator model is trained to modify or rerank the logits of the base model during decoding in order to satisfy the desired constraint (Yang & Klein, 2021), while preserving the distribution of the language model as much as possible.

Recently Deng & Raffel (2023) proposed the reward augmented decoding (RAD), an approach to train an autoregressive reward model as the discriminator. While RAD demonstrates high effectiveness for controlled generation, it scales poorly when the number of next token candidates grows, requiring a separate forward pass through the backbone of the reward model for each token candidate. In this aspect, RAD diverges from previous work of Liu et al. (2021) and Krause et al. (2021): the latter propose more efficient approaches using external attribute-conditioned language models, where each expert model only performs a single forward pass to predict the scores for all next token candidates.

In §3.1, we analyze RAD and reformulate its training objective in terms of approximating an incomplete reward matrix. We highlight that the RAD approach is flexible enough to represent a large space of reward matrices including those of high rank. However, when we empirically measure the rank of the reward matrix learned by RAD, it appears to be *low-rank*. This observation suggests that RAD might not use its full flexibility, which motivates us to reconsider the trade-off between efficiency and expressivity of reward models. By analyzing the incomplete reward matrix constructed

from the training data, we observe that it is enough to have a low-rank approximation to this matrix in order to reconstruct its observed values.

In light of this observation, we propose the *autoregressive reward model* (ARM), a low-rank reward model which combines the strengths of two paradigms: fast inference with language modeling prediction style and high quality of generations following the reward augmented decoding (RAD) approach (Deng & Raffel, 2023). We propose a simple strategy for how to transform a pretrained language model into an efficient autoregressive reward model. In the evaluation, we show that guided decoding with our model results in a comparable attribute control/fluency to the more flexible but more computationally intensive RAD approach.

## 2 PRELIMINARIES

### 2.1 GUIDED DECODING WITH EXTERNAL EXPERTS

In this section, we outline the approach of guiding a base language model with external token-level discriminators. At each step of decoding, both the base model and the discriminator observe an already generated prefix $x$, and cooperate to score the next token candidates $v \in V$. A language model predicts the logits $z_{\text{LM}}(\cdot|x) \in \mathbb{R}^{|V|}$ and the goal of discriminator is to augment these logits with reward scores $\hat{r}(\cdot|x) \in \mathbb{R}^{|V|}$. A standard practice is to consider only likely tokens $V' \subseteq V$ at each decoding step *e.g.* via top-$k$ (Fan et al., 2018; Deng & Raffel, 2023) or nucleus sampling (Holtzman et al., 2020):

$$z(v|x) = \begin{cases} z_{\text{LM}}(v|x) + \beta \hat{r}(v|x), & \text{if } v \in V' \\ -\infty, & \text{otherwise} \end{cases} \tag{1}$$

and the next token is sampled from the categorical distribution:

$$\tilde{p}(x) = \text{Softmax}(z(v|x)). \tag{2}$$

While some language models might have a restrictive application programming interface (API) for safety reasons, this line of work makes a reasonable assumption that we have access to the top-$k$ logits of a language model either directly or via API for a relatively small $k \ll |V|$.

To define reward scores, GeDi (Liu et al., 2021) and DExperts (Krause et al., 2021) use attribute-conditioned unidirectional language models (undesired attribute in GeDi or two LM experts for desired and undesired attribute in DExperts), trained via the standard language modeling objective on class-conditioned data: $\hat{r}_y(v|x) = z_t(v|x, y)$, where $y \in \{0; 1\}$ is the attribute (*e.g.* positive/negative sentiment). Given a prefix $x$ they only pass it once through the external language model backbone, relying on the linear output layer to obtain the scores for each of the next token candidates.

Alternatively, RAD (Deng & Raffel, 2023) trains a unidirectional reward model to predict the attribute of interest for a prefix *concatenated* with a next token candidate $\hat{r}_{\text{RAD}}([x, v])$, where $[\cdot, \cdot]$ denotes the concatenation of a prefix and a next token candidate. This approach requires passing each next token candidate as *input* to the model, thus, to obtain the scores for $k$ next token candidates $v$ for top-$k$ decoding RAD would need $k$ forward calls of the reward model, which can slow down inference significantly and constrains them to limit the number of next token candidates. Despite being less efficient, RAD outperforms the approach with attribute-conditioned language models in terms of controlled generation quality.

### 2.2 RAD TRAINING

In this section, we outline how RAD (Deng & Raffel, 2023) uses labeled data to train a reward model. At the training stage, we assume that we have a dataset $\mathcal{D} = \{(u^{(i)}, y^{(i)})\}_{i=1}^n$ of $n$ text utterances $u$ of length $l(u)$ and responses $y \in [0; 1]$. RAD trains a reward model to predict $y$ given a text input. While a simple strategy would be to train the reward model on full utterances $u$ from $\mathcal{D}$, it is important that the model predicts meaningful rewards also for *partially* generated utterances, needed during guided decoding. The RAD approach is to first extend the dataset by considering all partial utterance prefixes $u_{1:t}$ along with corresponding weights:

$$\mathcal{D}_f = \{(x, y, \lambda(t, l(u))) \mid x = u_{1:t}, t \in (1, \dots, l(u)), (u, y) \in \mathcal{D}\}, \tag{3}$$

where $\lambda$ are the weights $\lambda(t, l(u)) = \frac{t}{\sum_{t'=1}^{l(u)} t'}$ for each prefix used to up-weight prefixes closer to the full sentence, and $\sum_{t=1}^{l(u)} \lambda(t, l(u)) = 1$. Then, during training, RAD takes the input prefix $x = [x', v]$ and incurs a weighted squared loss for approximating the future reward:

$$\mathcal{L}(\hat{r}(v|x'), y, \lambda) = \lambda \cdot (\hat{r}(v|x') - y)^2. \tag{4}$$

During training, we can use teacher forcing to process all prefixes of an utterance in a single pass.

## 3 REWARD MODELING AS LOW-RANK MATRIX FACTORIZATION

### 3.1 ANALYSIS OF RAD

#### 3.1.1 REWARD MODELING AS MATRIX COMPLETION

To better understand the training objective of RAD, we start by looking at the optimization problems defined in §2.2, where we optimize a reward model to approximate the future responses. A unidirectional reward model can predict a reward value for each next next token candidate. If we enumerate all the contexts $x'$ in the training data and all possible next tokens $v$, we task a reward model to predict the values of $R \in \mathbb{R}^{N \times |V|}$, which we dub the *reward matrix*.

If each context would be observed only once, $R$ would have a single observed reward in each row. For short and common contexts we can observe more continuations per row, and also for some contexts there can be ambiguities: $\{(x, y_1, \lambda_1), ...(x, y_m, \lambda_m)\}$. From a mean squared error point of view, it is equivalent to compress these ambiguities by taking the weighted average of their $y$ (Appendix A):

$$R[x', v] = \frac{\sum_{\lambda, y \sim \mathcal{D}_f[x]} \lambda y}{\sum_{\lambda, y \sim \mathcal{D}_f[x]} \lambda}. \tag{5}$$

From this perpective, reward modeling can be interpreted as a matrix completion problem. The training dataset $\mathcal{D}_f$ gives us only an incomplete view of a true reward matrix $R$. Following the notation in the matrix completion literature (Mazumder et al., 2010), denote by $\Omega$ the set of indices of the observed entries $\{(x', v) \mid x = [x', v], x \in D_f\}$, and by $P_\Omega(R)$ the projection of $R$ that sets all indices outside $\Omega$ to zero. The full RAD objective is equivalent to minimizing $\|P_\Omega(R) - P_\Omega(\hat{R}_{\text{RAD}})\|_F^2$, where each entry $\hat{R}_{\text{RAD}}[x', v] = \hat{r}(v|x')$ can be computed with a forward pass.

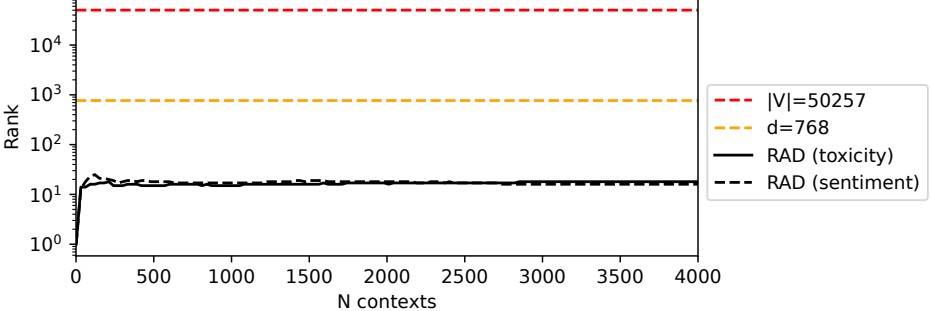

**Figure 1:** We numerically estimate the rank of $\hat{R}_{\text{RAD}}$ by increasing the number of seen randomly selected training prefixes (rows of the $\hat{R}_{RAD}$ matrix), and observe that the rank tends to be less than the model dimension $d = 764$ and much less than $|V|$, the maximal possible rank of $P_\Omega(R)$.

#### 3.1.2 RAD CAN BE HIGH-RANK, BUT IS NOT IN PRACTICE

Given a prefix $x$, RAD accepts a token candidate $v$ as an additional **input** to the model $\hat{R}_{\text{RAD}}[x', v] = \hat{r}_{\text{RAD}}([x', v])$, passing $v$ through the layers of the reward model. For this reason, we expect RAD to have the capacity to represent a large space of reward matrices including matrices with higher rank. In Appendix C.1, we empirically verify that **RAD is capable to approximate $P_\Omega(R)$ matrix with high rank**: $\text{rank}(P_\Omega(R)) > d$, where $d$ is the dimensionality of the model . This flexibility

does come at the cost: to score many next token candidates during top-$k$ decoding, RAD needs to do a forward pass through all layers of the model for each of the $k$ next token candidates. Hence, an important question is *do we need this flexibility at the cost of slower decoding?*

In Figure 1, we aim to measure the rank of $\hat{R}_{\text{RAD}}$ for RAD trained on two datasets: for detoxification and sentiment control tasks (discussed in detail in §5). To numerically estimate the rank, we follow Finlayson et al. (2024) and first sample $N$ random prefixes $x$ from the dataset $\mathcal{D}_f$ to calculate $N$ full rows of $\hat{R}_{\text{RAD}}$ (requiring $N \cdot |V|$ calls to the RAD reward model). Then we use singular value decomposition with the standard singular value cutoff to compute the rank (Appendix C.4). **We observe that the reward matrix learned by RAD tends to be *low-rank*,** suggesting that it is possible to use less flexible but faster reward models to improve the efficiency of reward models.

### 3.1.3 Is the training data full rank?

Note that the presence of a low-rank solution compatible with $\Omega$ does not imply that the true reward, if it could be fully observed, is necessarily low rank. We argue that low-rank predictions can partly be explained by the specifics of the training objective. Particularly, the incompleteness of $P_\Omega(R)$ makes it easier for a reward model to learn a low rank approximation. To understand why this is the case, consider a simple scenario, when all prefixes $x$ appear only once in the dataset. For this case, there exists a *rank-1* $\hat{R}$ compatible with $P_\Omega(R)$ (Appendix B.1).

To better understand this phenomenon, we would like to understand whether the $P_\Omega(R)$ can be fit with the low-rank model. We define the minimum rank of a partially observed matrix $\hat{R}$ as $\min \text{rank}(P_\Omega(R)) = \min\{\text{rank}(\hat{R}) : P_\Omega(R) = P_\Omega(\hat{R})\}$. We claim that the **data has low minimal rank.** Note that empirically calculating the minimal rank of the data is challenging due to the very large number of prefixes. We use a combination of theoretical and empirical approaches listed in Appendix B.2 to demonstrate that incomplete $P_\Omega(R)$ matrix can be fit with the low-rank matrix factorization with a small error (with the rank less than the model dimension).

### 3.2 Low-Rank Autoregressive Reward Model

Motivated to reduce the decoding costs of RAD, we propose ARM (Figure 2), a low-rank autoregressive reward model suitable for guided decoding, designed for efficient modeling of rewards scores for next token candidates. To ensure *prediction efficiency* of the reward model, we aim to revisit the language modeling style of prediction (Liu et al., 2021; Krause et al., 2021) and aim to predict the scores for all next token candidates with just a single forward pass through the backbone of a language model.

In contrast to RAD, ARM predicts the representation vector $h(x) \in \mathbb{R}^d$ given a prefix $x$ and uses output embeddings $e(v) \in \mathbb{R}^d$ to get the scores for all next token candidates. We propose the following ARM parametrization, similar to the how Dueling Network (Wang et al., 2016; Tang et al., 2023; Han et al., 2024) parametrizes the scores for the next tokens given the prefix:

$$\hat{r}_{\text{ARM}}(v|x) = \underbrace{\hat{r}_b(h(x))}_{\text{baseline}} + \Delta\hat{r}(e(v)|h(x)), \tag{6}$$

where the baseline predicts the score for the prefix $x$ and $\Delta\hat{r}$ predicts how observing a next token $v$ changes the score. Particularly, we use a *linear parametrization*:

$$\hat{r}_b(v|x) := \langle h(x), w \rangle \qquad \Delta\hat{r}(e(v)|h(x)) := \langle h(x), We(v) \rangle. \tag{7}$$

Here, we introduced two attribute-specific parameters: $w \in \mathbb{R}^d$ for modeling the baseline reward score of the prefix, and $W \in \mathbb{R}^{d \times d}$ to model marginal rewards for each next token candidate. Now it is clear that in contrast to RAD, ARM (as defined in Eq. (7)) performs a *low-rank matrix factorization* of $P_\Omega(R)$:

$$\hat{R}_{\text{ARM}} = H(w\mathbf{1}^T + WE) = HA, \tag{8}$$

where we stack all context representations $x'$ into $H \in \mathbb{R}^{N \times d}$ and all next token representations into $WE \in \mathbb{R}^{d \times |V|}$, and $\mathbf{1}$ is a column $d$-vector of all ones. rank inequality, $\text{rank}(\hat{R}_{\text{ARM}}) = \text{rank}(HA) \leq \min(\text{rank}(H), \text{rank}(A)) \leq d$, meaning that if $\text{rank}(P_\Omega(R)) > d$, ARM cannot possibly perfectly reconstruct $P_\Omega(R)$ no matter how flexible $h(x)$ is. In the language modeling literature the rank

bottleneck problem is known as the *softmax bottleneck* (Yang et al., 2018) and mitigation strategies are well-studied (Ganea et al., 2019; Chang & McCallum, 2022).

In the experiments (§5), we empirically demonstrate that our low-rank ARM can match the performance of the more flexible RAD on the two standard controlled generation benchmarks.

## 3.3 ARM TRAINING

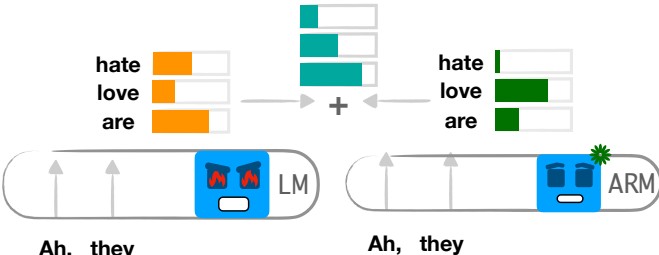

**Figure 2:** During decoding, we augment the logits of the base language model with reward scores from ARM. ARM uses the language model output embeddings to efficiently predict the rewards for next token candidates.

To train ARM, we rely on the RAD approach to train a reward model. We split $x$ into a last token and remaining prefix: $x = [x', v]$. We pass $x'$ as input to the model, and $v$ indexes output embeddings (7). We consider two types of experiments: training ARM on original responses from the dataset, and distillation experiment, where we train ARM to predict the scores of less efficient RAD.

For the first type of experiment , we train ARM on the responses from the dataset using the weighted squared loss:

$$\mathcal{L}(\hat{r}(v|x'), y, \lambda) = \lambda(\hat{r}(v|x') - y)^2 \tag{9}$$

For the second type of experiment , we train an ARM student to approximate the less efficient RAD teacher $\tilde{r}(x)$ (a frozen trained RAD) using the *distillation loss* (Hinton et al., 2015):

$$\mathcal{L}_{\text{dstl}}(\hat{r}(v|x'), \tilde{r}(x)) = (\hat{r}(v|x') - \tilde{r}(x))^2. \tag{10}$$

A reward model can only observe a limited number of next tokens $v$ given $x$ during finetuning. While the loss defined above provides a positive signal for some tokens $v$, it might be beneficial to regularize the prediction for other (unrelated) tokens, including rare or unseen tokens. In our parametrization (6), it is natural to push the predicted reward towards the baseline for unrelated tokens. We regularize the prediction of ARM to be close on average to the prefix baseline by forcing $\Delta\hat{r}$ to be close to 0 for randomly sampled token candidates:

$$\mathcal{L}_{\text{reg}}(h(x)) = \mathbb{E}_{v' \sim \text{Uniform}[V]} \left[ \Delta\hat{r}(e(v')|h(x)) \right]^2, \tag{11}$$

where we use one sample of $v'$ for each prefix position, sampling uniformly from the vocabulary. Particularly, a regularized model can learn to *abstain* by predicting the baseline score for each next token candidate, which will not change the distribution of a base model.

## 4 RELATED WORK

There are multiple approaches that investigate how to finetune a language model using attribute-conditioned data (desired/undesired examples). Keskar et al. (2019) finetunes a language model using control prompts. More recent approaches (Schulman et al., 2017; Stiennon et al., 2020; Lu et al., 2022) perform finetuning while regularizing the weights of the model to stay close to pretrained weights. Despite the efficiency of decoding, these methods might require more resources for finetuning if the language model is large, or might even be unusable if we only have access to the top-$k$ logits of base language model via an API.

Unlike finetuning, alternative approaches keep the language model untouched and use external models to guide the decoding from the base language model. Dathathri et al. (2019) use the gradients

from a discriminator to modify the prefix activations of the base model during decoding. However, gradient-based methods are costly to use during decoding since they require backpropagating through the large base model.

Closest to our work are *gradient-free* guided decoding methods, where we have access to the frozen base language model and use external models to guide the sampling process from the base model. GeDi (Krause et al., 2021) proposes to use class-conditioned language models as discriminators to augment the decoding and efficiently compute the scores for next token candidates. DExperts (Liu et al., 2021) improves the quality of GeDi introducing an ensemble of two class-conditioned language models finetuned on desired and undesired data. We look closer into parametrization of the output layer and propose to decouple the prefix score from marginal scores of the next token candidates. Additionally, for our parametrization, we propose a regularization, which makes it easier for the model to abstain.

More recently, Deng & Raffel (2023) and Sitdikov et al. (2022) argue to use discriminator models Yang & Klein (2021) to guide the decoding. Sitdikov et al. (2022); Dekoninck et al. (2023) use available *bidirectional* Transformers to guide the base language model, which, however, requires to recompute all prefix tokens at each decoding step. To tackle this issue, RAD (Deng & Raffel, 2023) proposes a *unidirectional* model suitable for caching of prefix activations. They train a reward model on partial prefixes to predict the expected future attribute and demonstrate high quality of controlled generation.

In our work, we focus on the analysis of the RAD, while an alternative but related direction follows reinforcement learning (RL) approach. Particularly, RAD uses value-function style parameterization, while ARM is Q-function style. Among RL-based approaches, Mudgal et al. (2024), Chakraborty et al. (2024) parametrize the value function, which results in higher decoding complexity; Cao et al. (2022) parametrize the Q-function, resulting in similar efficiency to ARM. To the best of our knowledge, there is little attention to the implied efficiency-quality trade-off that we study in our work. The closest to our analysis is the recent work of Han et al. (2024), where they compare both parametrizations in relation to language modeling, however they observe that value function parametrization outperforms Q-function parametrization, which disagrees with our work.

To summarize, we complement the previous work, by zooming in into the parametrization of an autoregressive reward model. We highlight the trade-off between efficiency and expressiveness of a reward model, and showcase that, for tasks and datasets we consider, higher rank-expressiveness can be traded for higher efficiency without quality drop. We hope our analysis will inform future work on the design choices of autoregressive reward models.

## 5 EXPERIMENTS

### 5.1 CONTROLLED GENERATION

We follow previous work (Deng & Raffel, 2023; Liu et al., 2021) and evaluate ARM on two controlled generation tasks: detoxification and sentiment control.

In our experiments, we guide the decoding from a base model using a smaller finetuned reward model with the same tokenizer. Namely, we guide GPT-2-Large using a reward model finetuned from GPT-2-Small, and we guide the LLaMa-2-(7b/13b) (Touvron et al., 2023) base language model with a reward model finetuned from TinyLLaMa (Zhang et al., 2024). We finetune all parameters of the reward models except input/output embeddings, which remain frozen (we hope that, this way, the reward model generalizes better to unseen tokens).

We conduct experiments in two regimes: first, by distilling less efficient RAD (Deng & Raffel, 2023) using $\mathcal{L}_{\text{dstl}}$ loss (10); second, by training a reward model from scratch on the responses from the datasets using cumulative loss $\mathcal{L}$ (9). In both settings, we use additional regularization $\mathcal{L}_{\text{reg}}$ by default. For evaluation, we perform guided decoding using *top-k* sampling from the categorical distribution defined in (2), where *top-k* candidates are selected taking $k$ largest logits of the base model at the current decoding step.

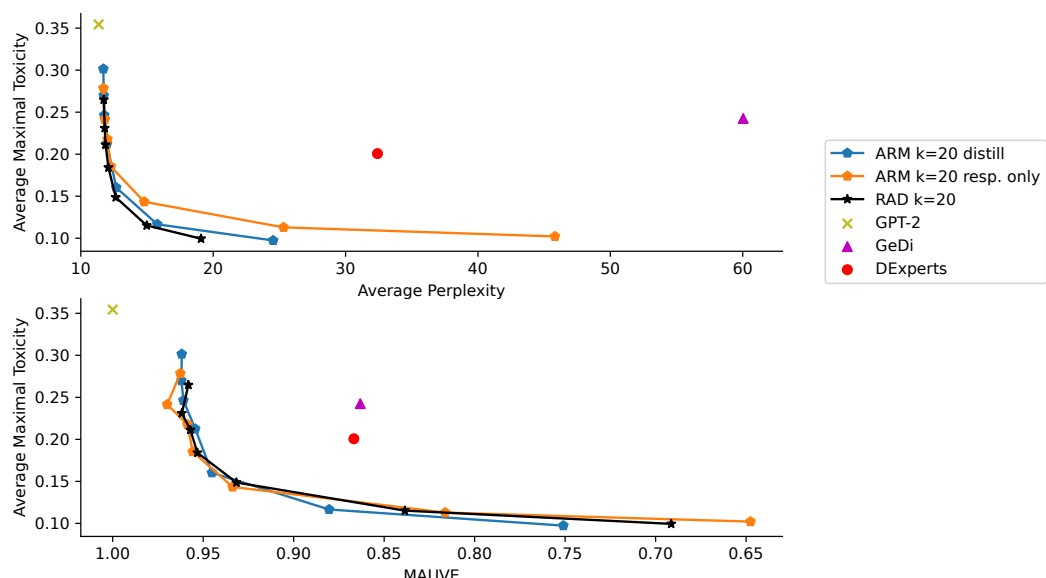

**Figure 3:** ARM student (distil) shows comparable toxicity/fluency trade-off with the teacher RAD, where the ARM student closely matches the performance of the teacher RAD. ARM trained on original responses (ARM resp. only) shows slightly worse fluency and similar toxicity level. We rerun the evaluation for RAD, GeDi and DExperts with an up-to-date Perspective API classifier. We include the results with other baselines from Deng & Raffel (2023) in Figure 12 (see Appendix F.1.1).

## 5.2 DETOXIFICATION

For the detoxification evaluation, we follow previous work (Deng & Raffel, 2023; Liu et al., 2021) and evaluate samples from guided decoding given a 10k subset (Liu et al., 2021) of prompts from the RealToxicityPrompts dataset (Gehman et al., 2020). We follow Deng & Raffel (2023) and Liu et al. (2021) and finetune our model on 2M pairs of text and continuous 'toxicity' responses between 0 and 1 from the Jigsaw Unintended Bias in Toxicity Classification challenge (cjadams et al., 2019). Like previous work, we train our model on 7 independent responses ('toxicity', 'severe toxicity', 'obscene', 'identity attack', 'insult', 'threat', 'sexual explicit') with different head parameters $w_i, W_i, i \in \{1, ..., 7\}$ for each sub-task. During decoding, we only use the 'toxicity' predictor. For the distillation experiment, we use the same dataset, and the released toxicity discriminator from Deng & Raffel (2023) as a teacher.

During decoding, we sample 25 continuations generating at most 20 new tokens. To evaluate toxicity, we use an external closed-source toxicity classifier *Perspective API* (Lees et al., 2022), and following previous work (Deng & Raffel, 2023; Liu et al., 2021), we rely on the *Maximal Average Toxicity* metric, which is the maximal toxicity score value over 25 samples for a given prompt, averaged over the set of 10k prompts. We also report *Toxic Rate*, which is calculated as the probability that at least one out of 25 continuations is toxic according to Perspective API (toxicity score > 0.5); and *Diversity* score, which is the average number of distinct $n$-grams normalized by the length of text (Li et al., 2018). To evaluate the fluency of model generations, we follow previous work (Liu et al., 2021; Deng & Raffel, 2023) and report the average perplexity of the GPT-2-XL when generating from the GPT-2-Large model; and the OLMo[1] to evaluate the LLaMa family as in Lovelace et al. (2024). As an additional fluency metric, we report MAUVE (Pillutla et al., 2021) to measure the distance between unguided and guided generations (details in Appendix E). In the experiments, we will look at the toxicity/fluency trade-off, alternating the weight $\beta$ of the discriminator (see Table 2 and Table 3). We expect to obtain a model with both low toxicity according to the Perspective API, and high fluency.

---

[1] https://huggingface.co/allenai/OLMo-1B

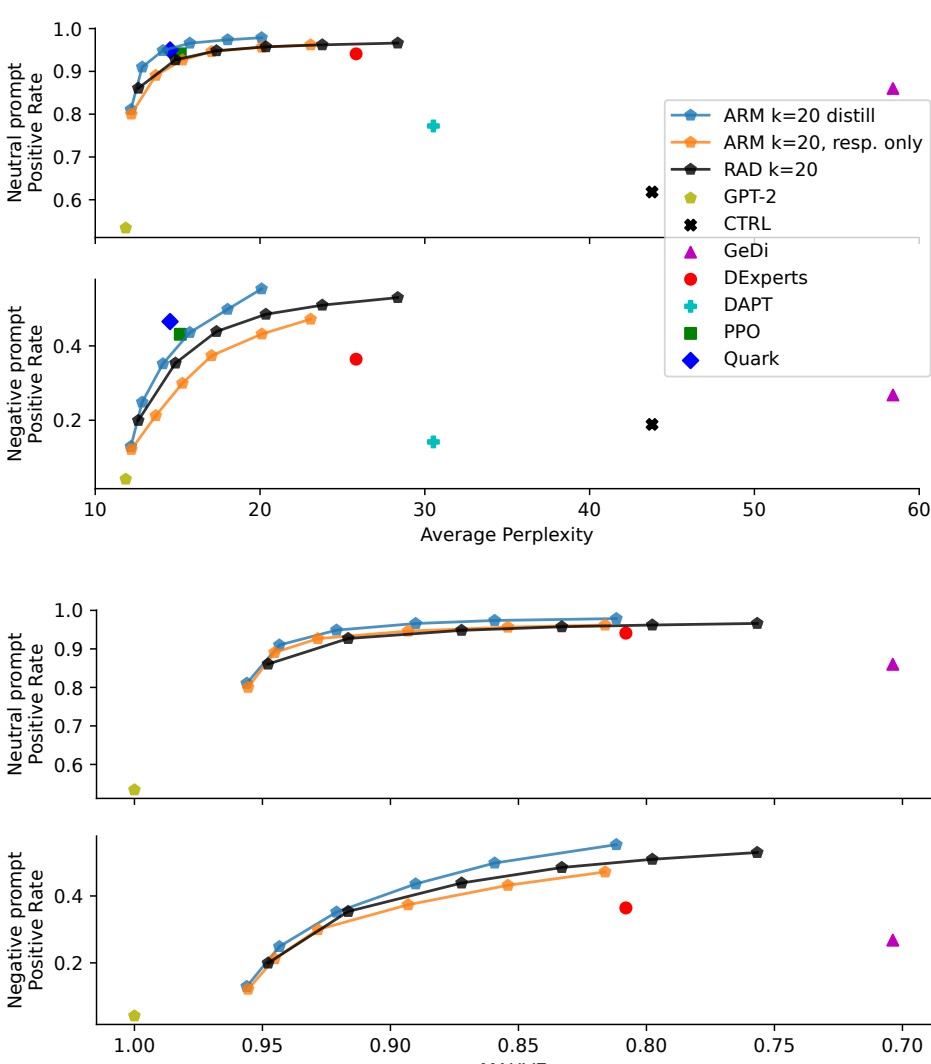

**Figure 4:** For the sentiment control task, ARM trained on responses only lags slightly behind the RAD baseline, while student ARM outperforms the teacher RAD model. For the plot with average perplexity, we include the results from Deng & Raffel (2023) for other baselines for reference.

Since toxicity scores from the Perspective API can change overtime, which can complicate the evaluation, in Appendix F.3.1 we evaluate our detoxification models with an open-weight toxicity classifier,[2] where we observe the same relative results as with Perspective API scores.

## 5.3 SENTIMENT CONTROL

For sentiment control, we follow previous work (Li et al., 2018; Sudhakar et al., 2019; Liu et al., 2021; Deng & Raffel, 2023) to evaluate the samples given a prompt from one of the three categories: $2.5K$ *negative*, $5K$ *neutral*, and $2.5K$ *positive* prompts from OpenWebText (Gokaslan & Cohen, 2019). To finetune ARM on responses only, we follow Deng & Raffel (2023) and finetune our model on millions of reviews from the Amazon Polarity (Zhang et al., 2015) and SST-2 (Socher et al., 2013) datasets. To distil the sentiment discriminator of Deng & Raffel (2023), we use text examples from the Amazon Polarity dataset. Additional training details are provided in Appendix D.

---

[2]https://huggingface.co/nicholasKluge/ToxicityModel

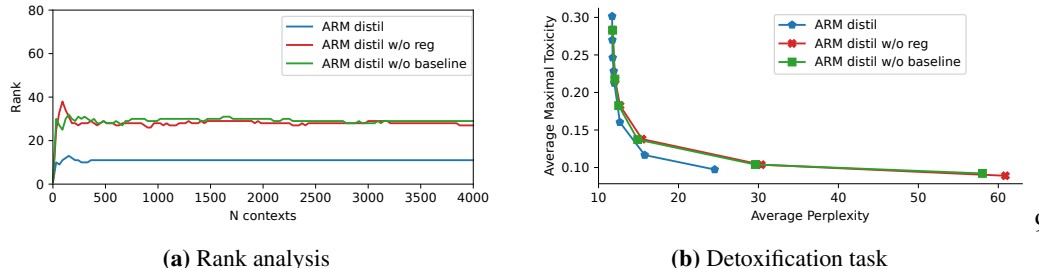

**(a)** Rank analysis

**(b)** Detoxification task

**Figure 5:** Ablation experiment for distilled ARM, on the detoxification task with top-$k$=20. On the right, we observe that regularization towards the baseline results in better fluency of generated samples. On the left, we observe that regularization lowers the rank of the model's outputs $\mathrm{rank}(\hat{R}_{\mathrm{ARM}})$.

For evaluation, we follow Deng & Raffel (2023), and use the average *Positive Rate* metric *w.r.t.* the finetuned DistilBERT classifier (Sanh et al., 2019) provided via the HuggingFace.[3] As in the toxicity task, we use GPT-2-XL/OLMo and MAUVE to evaluate the fluency of the sampled continuations, and we expect to obtain a high Positive Rate and high fluency.

### 5.4 RESULTS

To compare RAD and ARM, we rely on the methodology of Deng & Raffel (2023) and Liu et al. (2021), and visualize the trade-off plots for both models varying the control parameter $\beta$. Namely, each point in the figure will represent two metrics: toxicity/sentiment along the vertical axis and fluency along the horizontal axis. From this plot, we can read e.g. what fluency (perplexity/mauve) can be achieved for a given 'target' toxicity. To compare two models, we compare their curves (in the same plot). Our hypothesis is that ARM will perform similar to the more flexible RAD approach, meaning that the trade-off plots for these models will be close to each other.

**Detoxification.** For the detoxification task (Figure 3), our efficient student (ARM) closely follows the RAD teacher for toxicity control/fluency trade-off. We observe that ARM trained on responses only shows slightly worse fluency *w.r.t.* average perplexity for lower levels of toxicity. For completeness, in Figure 12, we include the results for other baselines from Deng & Raffel (2023) computed for an older version of Perspective API. For guided decoding from the LLaMa-2-(7b/13b), we observe that again ARM closely follows RAD in terms of toxicity/fluency trade-off (see Figure 14 in Appendix F.1.1).

**Sentiment control.** From the results on the sentiment control task in Figure 4, we observe that the ARM student model shows slightly better trade-off than the RAD teacher model, closely following approaches that require training using feedback from the evaluation pipeline (Lu et al., 2022, Quark), (Stiennon et al., 2020, PPO). Again, ARM trained on original responses slightly lags behind but still performs competitively compared to other guided decoding baselines.

**Summary.** Our empirical results suggest that ARM can match the quality of more flexible but less efficient RAD. We observe that distilling the RAD teacher into the ARM student results in slightly higher quality compared to training ARM on original responses. One difference is that when training from data, we will see short contexts multiple times with different reward responses and must implicitly converge to their average, while in distillation, the teacher already performs this compression and provides a single deterministic target $\hat{r}(v|x)$ for every context $(x, v)$. We conjecture that this may lead to better-trained distilled models.

### 5.5 ABLATION

In this section, we investigate the effect of adding the baseline component Eq. (7) and of regularization Eq. (11). In Figure 5, we experiment with the distilled version of ARM and observe that turning off

---

[3]https://shorturl.at/9MqDp

| Model | N calls |
|---|---|
| GeDi (Krause et al., 2021) | 1 |
| DExperts (Liu et al., 2021) | 2 |
| RAD (Deng & Raffel, 2023) | $k$ |
| ARM (Ours) | 1 |

**Table 1:** Number of input tokens a discriminator model needs to process for a single decoding step with $k$ next token candidates. All included models are based on the unidirectional Transformer (Vaswani et al., 2017) and support the caching of prefix activations.

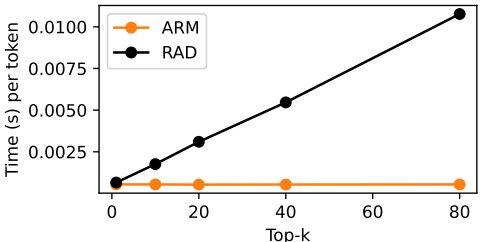

**Figure 6:** RAD processes the $k$ next token candidates separately as input requiring more time compared to ARM, which relies on the output layer to obtain the scores for all next tokens.

regularization, or further removing the baseline from the parametrization results in still adequate but slightly worse fluency as measured by perplexity, and a comparable toxicity decrease. By further analyzing the ranks of $R_{\mathrm{ARM}}$ with and without regularization, we observe that regularization effectively decreases the rank of $R_{\mathrm{ARM}}$ which might explain the higher fluency of regularized model. Particularly, a very strong regularization would result in the model always predicting the baseline score for each of the next tokens (corresponds to the rank-1 output), which does not modify the original distribution of the model (the best fluency).

## 5.6 EFFICIENCY

We consider using a reward model to compute the scores for $k$ candidate tokens at each of $L$ steps of decoding. Similar to RAD (Deng & Raffel, 2023), ARM is based on the *unidirectional* Transformer architecture (Vaswani et al., 2017), which means that we can cache the prefix activations during decoding. To compute the prediction for $k$ next token candidates $v$ given a prefix $x$, RAD needs to pass $k$ next tokens as *input* to the Transformer model, thus RAD processes $O(Lk)$ tokens during decoding. In contrast, ARM only processes $O(L)$ tokens as input to the Transformer model and relies on the output layer to efficiently compute the scores for all next token candidates. In Table 1, we summarize how many tokens external expert models process during top-$k$ decoding. In Figure 6, we measure the time per generated token when running the decoding for the toxicity task with ARM and RAD (Deng & Raffel, 2023) on a single RTX A6000 GPU.

## 6 CONCLUSION

We review the recently proposed RAD approach of training a reward model for the guided decoding, and we reformulate it as the incomplete reward matrix learning problem. In the light of the rank analysis of the reward matrix, we observe that the high flexibility of RAD might not overweight its lower efficiency during decoding. We present the low-rank ARM, an efficient approach to parameterize the reward model, suitable for autoregressive decoding, caching of prefix activations, and prediction of next token scores with a single call of a backbone model. We bridge the gap between two paradigms of training external expert models, demonstrating that we can have both efficient and effective controlled generation.

## LIMITATIONS

The models discussed in this work can only reduce the probability of generating the toxic responses, not prevent it. Moreover, evaluation of toxicity is far from perfect, and even a very low toxicity score from automatic evaluation such as Perspective API does not necessary mean that the sample is 'safe'. Furthermore, we should not exclusively rely on toxicity when evaluating the safety of samples from language models due to the complexity and variability of language. It is also not clear that by reducing toxicity, we are not introducing other harms. Furthermore, both RAD and our models represent low-rank $\hat{R}$ and further qualitative research is needed to investigate whether certain toxicity patterns require high rank to represent them.

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

SUPPLEMENTARY MATERIAL

## A  REWARD MATRIX

To train a reward model, we use weighted mean squared loss, for which the weighted mean recovers the minimum:

$$r^* = \arg\min_r \sum_{\lambda,y} \lambda(r-y)^2 = \frac{\sum_{\lambda,y} \lambda y}{\sum_{\lambda,y} \lambda} \tag{12}$$

*Proof.* $\frac{\partial}{\partial r} \sum_{\lambda,y} \lambda(r-y)^2 = \sum_{\lambda,y} \frac{\partial}{\partial r}[\lambda(r-y)^2] = \sum_{\lambda,y} 2\lambda(r-y) = 2(r\sum_{\lambda,y}[\lambda] - \sum_{\lambda,y}[\lambda y]) = 0$. Hence, $r^* = \frac{\sum_{\lambda,y} \lambda y}{\sum_{\lambda,y} \lambda}$  □

## B  FACTORIZATION OF $P_\Omega(R)$

Any matrix $R \in \mathbb{R}^{N \times |V|}$ can be factored as $R = UV^T$ with $U, V$ of dimensions $N \times q$; $|V| \times q$. If $R$ is *incomplete*, then there are in general multiple possible factorizations of $P_\Omega(R)$ compatible with the observed values.

### B.1  RANK-1 CASE

To get better intuition why the incompleteness of $P_\Omega(R)$ allows to find a compatible factorization with lower minimal rank, consider a simple example. If we only know 1 element per row of $R$, then minimal rank of $P_\Omega(R)$ is equal to 1. To prove this, consider completing $P_\Omega(R)$ such that each row is filled with the same element (the only one known for this row):

$$\begin{pmatrix} 1 & ? & ? \\ ? & 4 & ? \\ ? & ? & 3 \end{pmatrix} \rightarrow \begin{pmatrix} 1 & 1 & 1 \\ 4 & 4 & 4 \\ 3 & 3 & 3 \end{pmatrix}$$

### B.2  ESTIMATING THE MINIMAL RANK OF THE DATA

Empirically calculating minimal rank is challenging due to the very large number of prefixes (row of the matrix), particularly, the large portion of the prefixes have unique continuation. We show how we simplify the minimal rank estimation by considering only the prefixes with two or more continuations, and demonstrate that partially observed $\hat{R}$ can be fit with a low rank matrix factorization.

**Lemma 1.** *Consider a partially observed reward (sub-)matrix $P_\Omega(R)$, such that for every row, only one element of $R$ is observed. Then there exists $\hat{R}$, such that $\operatorname{rank}(\hat{R}) = 1$ and $\|P_\Omega(R) - P_\Omega(\hat{R})\|_F^2 = 0$.*

*Proof.* Let $v_i$ be the observed value in $R$ at row $i$, and let $\hat{R} = v\mathbf{1}^\top$; i.e., we complete the whole row with the same value. This rank-1 matrix achieves zero loss.  □

Given a training dataset of responses and text utterances, there will be many unique prefixes, for which the Lemma 1 is applicable.

**Lemma 2.** *Let $P_\Omega(R)$ be a partially observed matrix and $[P_\Omega(R)]_2$ correspond to the submatrix formed only with the rows of $P_\Omega(\hat{R})$ with at least two observed indices. Then, $\min \operatorname{rank}(P_\Omega(R)) \le 1 + \min \operatorname{rank}([P_\Omega(R)]_2)$*

Next, we demonstrate that $[P_\Omega(R)]_2$ can be fitted by a model that produces low rank $\hat{R}$: $\min \text{rank}([P_\Omega(R)]_2) \le d - 1$ for our specific dataset, and using Lemmas 1 and 2, we conclude that $\min \text{rank}(P_\Omega(R)) < d$. This implies that the training dataset $D_f$ can be fit by a reward model that produces low-rank $\hat{R}$, regardless of the specifics of said model.

Here we check that there exists a factorization of $[P_\Omega(R)]_2$ with rank $q$ at most 512 ($q$ is less than the model dimension $d = 764$). In general, finding minimal rank factorization of incomplete matrices is known to be NP-hard, and usually convex relaxation such as minimization of the nuclear norm is considered (see Nan (2009)). To factorize $P_\Omega(R)$, we use the alternating least squares algorithm (Mazumder et al., 2010; Hastie et al., 2015).[4] To accelerate convergence to a good solution, we first optimize for 50 iterations with a trace norm penalty of $\lambda = 0.01$ (*i.e* we start by solving a non-convex approximation of a convex problem) followed by an additional 50 iterations with no trace penalty. At the end, the mean squared error (MSE) over the observed entries is 0.00056. Given the large scale of $P_\Omega(R)$, it is possible that a better fit could be found nevertheless we find this sufficient evidence to claim that low-rank models could indeed fit the training data.

## C  $\hat{R}_{\text{RAD}}$ AND $\hat{R}_{ARM}$

### C.1  $\hat{R}_{\text{RAD}}$

In this experiment, we empirically verify that RAD is capable to approximate $P_\Omega(R)$ matrix with $\text{rank}(P_\Omega(R)) > d$, where $d$ is the dimensionality of the model. We finetune RAD initialized from the GPT-2-Small (with $d = 764$) on a synthetic data constructed as follows. We generate $R_I = I(n), n = 1024 > d$, an identity matrix of size 1024.

With $R_I$ as a full rank $1024 \times 1024$ submatrix of $P_\Omega(R)$, $\text{rank}(P_\Omega(R)) > d$. We verify that we can train RAD to fit this train matrix obtaining the MSE $< 10^{-7}$.

### C.2  $\hat{R}_{ARM}$

ARM approximates $P_\Omega(R)$ as a product of two rank $d$ matrices, hence for ARM, the lowest MSE for the synthetic experiment from the previous section is recovered for rank-$k$ singular value decomposition of $R_I$, which is $I(k)$. Hence for ARM, MSE $\ge (n - d)/n^2 = 0.00024$.

We thus conclude that RAD (in contrast to ARM) is indeed capable of representing $P_\Omega(R)$ matrices with a rank higher than $d$.

### C.3  REAL DATA EXPERIMENTS

For the experiment with the real datasets for the detoxification and sentiment control tasks, in Figure 7, we numerically measure the rank of $R_{RAD}$ and $R_{ARM}$, and observe that both ARM and RAD learn low-rank reward matrices. We thus conclude that both these models have needed capacity to represent the incomplete $P_\Omega(R)$ matrices obtained from the datasets.

### C.4  NUMERICAL RANK

To compute rank of $n \times m$ matrix, we use the default cutoff in Numpy[5] and PyTorch[6] at the time of writing, which is to say we count only singular values above $\max(m, n)\varepsilon\sigma_1$, where $\varepsilon$ is the machine epsilon for the corresponding data type, i.e., the difference between 1.0 and the next smallest representable number larger than 1.0, and $\sigma_1$ is the largest singular value .

There are potential issues that may arise when computing the numerical rank. One issue is that the singular values, especially for the matrices coming from 32bit float precision neural network, will not be exactly zero, so this is why libraries like Numpy or PyTorch use a precision-based cutoff

---

[4]https://cran.r-project.org/web/packages/softImpute/index.html
[5]https://numpy.org/doc/stable/reference/generated/numpy.linalg.matrix_rank.html
[6]https://pytorch.org/docs/stable/generated/torch.linalg.matrix_rank.html

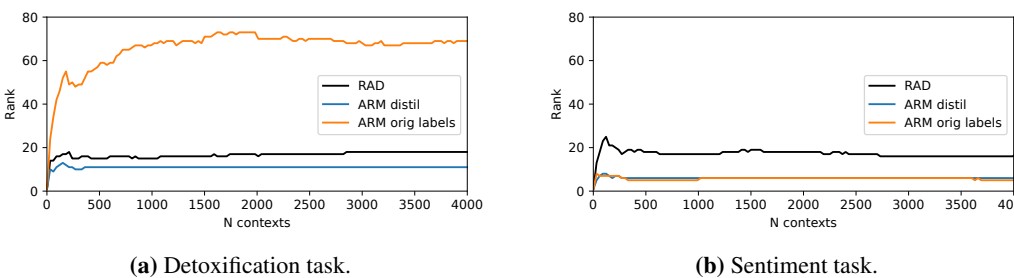

**(a)** Detoxification task.    **(b)** Sentiment task.

**Figure 7:** We numerically estimate the ranks of both $\hat{R}_{\text{RAD}}$ and $\hat{R}_{\text{ARM}}$ increasing the number of training prefixes (rows of $\hat{R}$). In all cases, the ranks tend to be less than the model dimension $d = 764$. This means that rank-capacity of ARM is sufficient to capture the training datasets for the detoxification and sentiment tasks.

for singular values that should be considered indistinguishable from zero; we use the default such parameters. The other issue is that the number of rows in the reward matrices is very high and we follow the work of Finlayson et al. (2024) and estimate rank by sampling rows from the matrix. Different submatrices can have different ranks, but we sample i.i.d. to prevent this.

## D    TRAINING DETAILS

To train reward models, we reuse the hyperparameters from Deng & Raffel (2023), where possible. We finetune the reward models with Adam optimizer (Kingma & Ba, 2015) with $\beta_1 = 0.9, \beta_2 = 0.95, \epsilon = 1\text{e}{-}12$. We use weight decay 0.02, and batch size 100.

To train ARM, we initialize the parameters with the pretrained GPT-2-Small/TinyLLaMa[7] weights, and freeze the shared input-output embedding parameters. Alternative strategy would be to use parameter efficient finetuning (Hu et al., 2022; Sidahmed et al., 2024) .

## E    MAUVE

To complement perplexity as a measure of fluency, we use MAUVE (Pillutla et al., 2021) as one of the fluency metrics. For reference texts, we take the generations of unguided model (GPT-2, or LLaMa-2-(7b/13b). Thus, this metric should capture how close the distribution of the continuations of a guided model is to the distribution of the original language model. To calculate MAUVE, we follow recommendations of He et al. (2023) and use ELECTRA-large model to obtain the text representations. We use the hyperparameters of Pillutla et al. (2021): $c = 5$ for the scaling constant; $k-$means for the quantization algorithm with 500 iterations, and $n/10$ clusters where $n$ is the number of generations. To compute MAUVE, we use 1000 prompts from the evaluation dataset.

### E.1    DETOXIFICATION

For the detoxification task, we finetune ARM with the learning rate $10^{-5}$ for 5 epochs.

For the LLaMa-2, we additionally finetune RAD with the TinyLLaMa backbone for the fair comparison with ARM.

### E.2    SENTIMENT CONTROL

To finetune ARM on responses only for sentiment control task, we first finetune the model with the learning rate $10^{-5}$ on the Amazon Polarity dataset, and then finetune it for 5 epochs on the SST-2 dataset with the learning rate $2\text{e}{-}6$. For distillation experiment, we finetune ARM for 5 epochs with the learning rate $10^{-5}$ on Amazon Polarity dataset.

---

[7]https://huggingface.co/TinyLlama/TinyLlama-1.1B-intermediate-step-1431k-3T

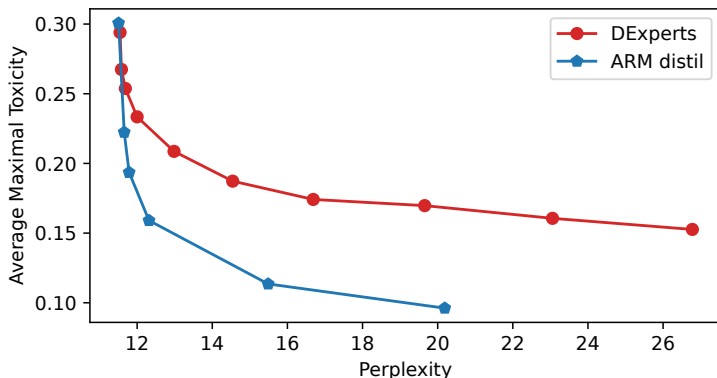

**Figure 8:** Comparison of toxicity/fluency trade-off between ARM (distil) and DExperts. We rerun the sampling from these two models using *top-k* decoding with $k = 20$. Results are calculated over randomly selected 1000 prompts. We observe, that ARM show better constraint satisfaction/fluency than DExperts.

## F    RESULTS

### F.1    DETOXIFICATION

#### F.1.1    RESULTS WITH PERSPECTIVE API CLASSIFIER

In this section, we report full results with the Perspective API as a toxicity classifier.

**GPT-2.**    Results for the detoxification task with the GPT-2-Large base model and GPT-2-small reward model, are presented in Table 2.

We present the results for ARM and RAD with *top-k* decoding with $k = 40$ in Figure 11. We observe similar relative performance of ARM compared to RAD as in the experiment with $k = 20$, presented in the Figure 3.

**LLaMa-2.**    Results for detoxification task with LLaMa-2-(7b/13b) base model and TinyLLaMa reward model are presented in Figure 14 and Table 4.

**Baselines.**    Additionally, in Figure 12, we include results from Deng & Raffel (2023) for other baseline models (for an older version of Perspective API).

To highlight the difference between the ARM and DExperts, we show the trade-off plot for DExperts model in Figure 8, varying the $\alpha$ scalar parameter for DExperts. As we can observe, the ARM has better constraint satisfaction / fluency trade-off than DExperts model. We attribute this to the difference in the training objectives of the expert models (reward modeling or language modeling), as argued in (Deng & Raffel, 2023).

### F.2    ADDITIONAL ABLATION RESULTS

#### F.2.1    LOSS CHOICE

We perform an ablation study for the choice of the loss function used to train ARM Eq. (9). There, we follow the approach of (Deng & Raffel, 2023), where they introduce the squared loss (see section 2.1 Unidirectional Reward Model). An alternative strategy would be to use the binary cross-entropy loss, using the fact that for our datasets responses $y$ are from $[0; 1]$ range:

$$\mathcal{L}_{ce}(\hat{r}(v|x'), y, \lambda) = \lambda(y \log \sigma(\hat{r}(v|x')) + (1 - y) \log(1 - \sigma(\hat{r}(v|x')))), \tag{13}$$

where we introduced $\sigma(x) = 1/(1 + e^{-x})$ function to softly map the predictions of ARM into $[0; 1]$ range, which is also used during generation.

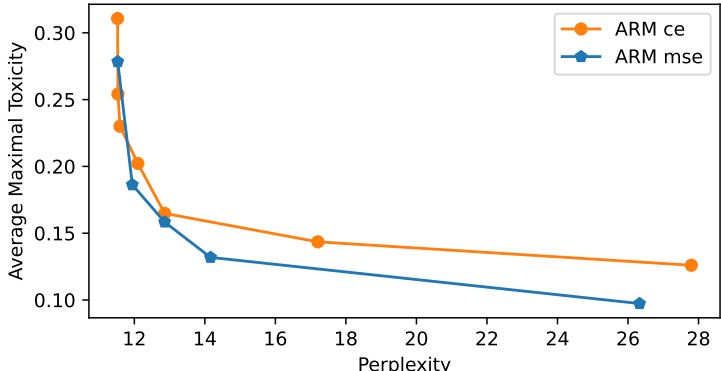

**Figure 9:** Comparison of ARM model trained on original responses with squared loss vs with cross-entropy loss. We rerun the sampling from these two models using *top-k* decoding with $k = 20$. Results are calculated over randomly selected 1000 prompts. We observe, that ARM trained with the squared loss show slightly better constraint satisfaction/fluency than ARM trained with cross-entropy loss.

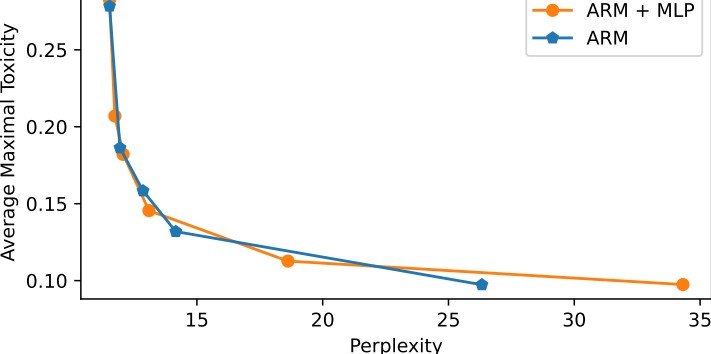

**Figure 10:** Comparison of ARM trained on original responses with linear parametrization vs with non-linear MLP parametrization. We rerun the sampling from these two models using *top-k* decoding with $k = 20$. Results are calculated over randomly selected 1000 prompts. We observe, that both parametrizations perform very closely.

Figure 9 demonstrates that the ARM trained with the squared loss slightly outperform the ARM trained with the binary cross-entropy loss.

### F.2.2 MLP VS LINEAR PARAMETRIZATION

In this ablation, we consider replacing the linear parametrization of ARM Eq. (8) with a non-linear MLP parametrization:

$$\Delta \hat{r}_{\text{ARM+MLP}}(x) := W_1 \sigma(W_2 E^T W^T h(x)^T), \tag{14}$$

where $W_1 \in \mathbb{R}^{d \times |V|}$; $W_2 \in \mathbb{R}^{|V| \times d}$. As we observe in Figure 10, MLP parametrization performs on par with the linear parametrization. We thus recommend using a more simple linear parametrization.

### F.3 SENTIMENT CONTROL

Here, in Figure 13, we include the additional results for the RAD and ARM with *top-k* decoding with $k = 40$.

### F.3.1 RESULTS WITH ROBERTA CLASSIFIER

In addition to toxicity scores with Perspective API, we provide the results with the open-weight RoBERTa toxicity classifier (Corrêa, 2023) for the guided generation with GPT-2 (Figure 15 and Table 5) and the LLaMa-2 (Figure 16 and Table 6). We notice that results for the average maximal toxicity with RoBERTa are relatively similar to the result with Perspective API. We hope that with an open-weight classifier it will be easier for the community to directly compare to the published results without the need to recompute the API scores.

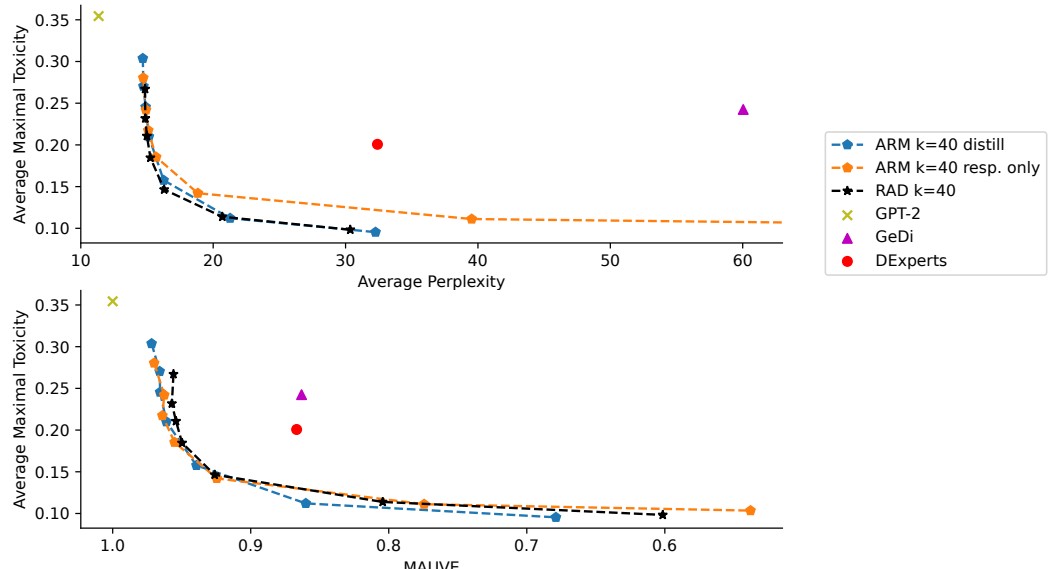

**Figure 11:** Additional results for detoxification task for ARM and RAD with $k = 40$.

### F.4 SENTIMENT CONTROL

Results for sentiment control task with the GPT-2-Large are presented in Table 3.

### F.5 GENERATED EXAMPLES

Examples for the detoxification and sentiment control are presented in the Table 7, Table 8 and Table 9.

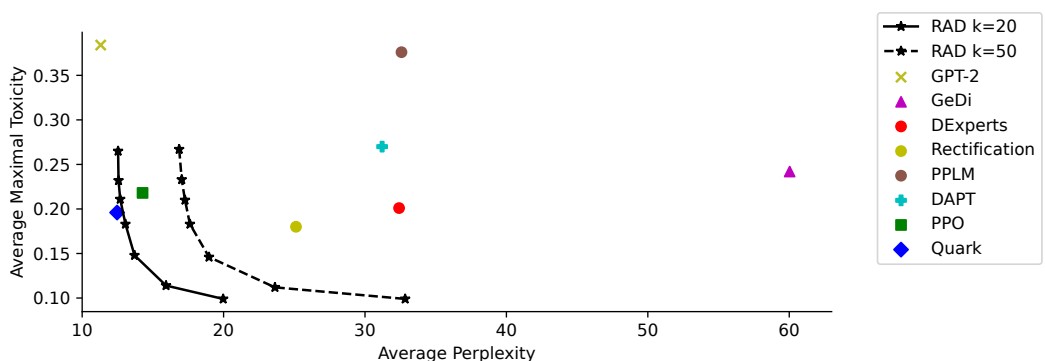

**Figure 12:** Detoxification results reported in Deng & Raffel (2023) with Perspective API with GPT-2-Large model (API queries made between May and June 2023).

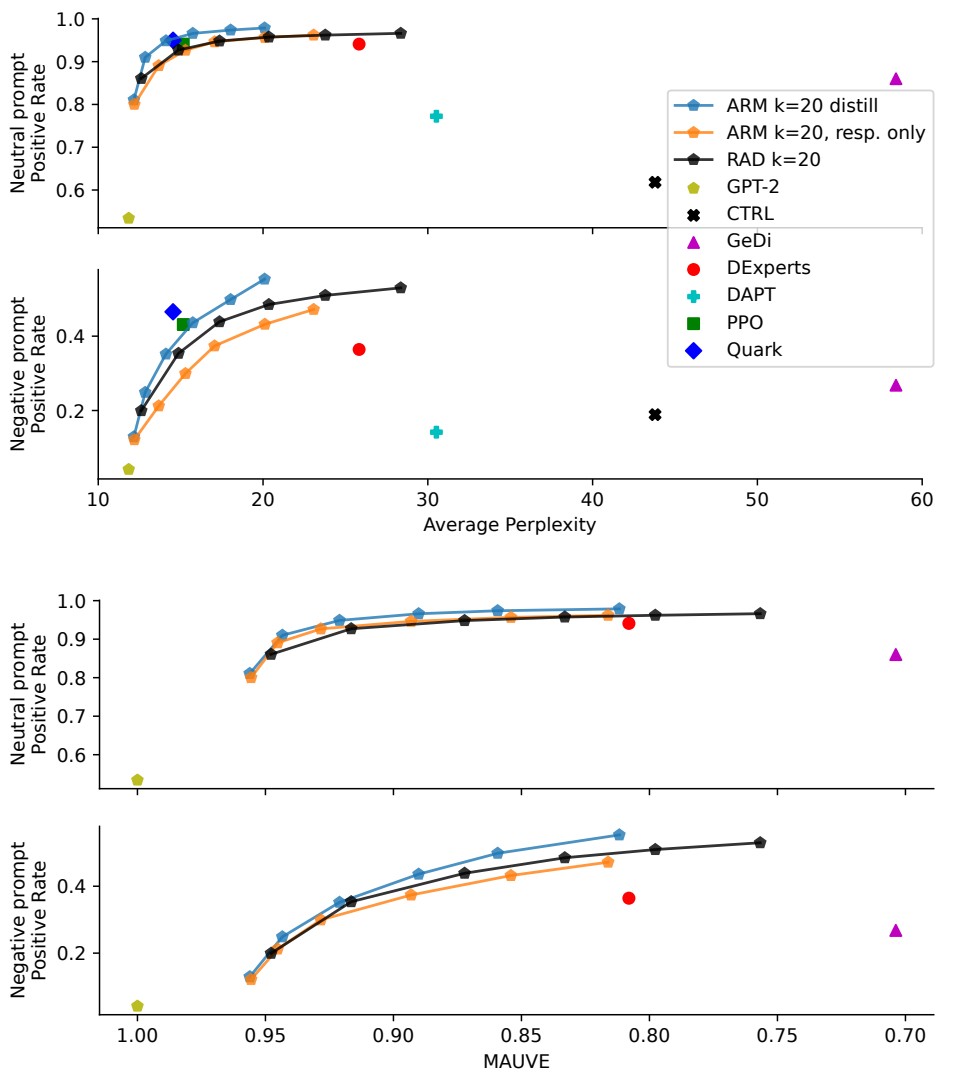

**Figure 13:** Additional results for sentiment control task with $k = 40$. For this plot with average perplexity, we include the results from Deng & Raffel (2023) for other baselines for reference.

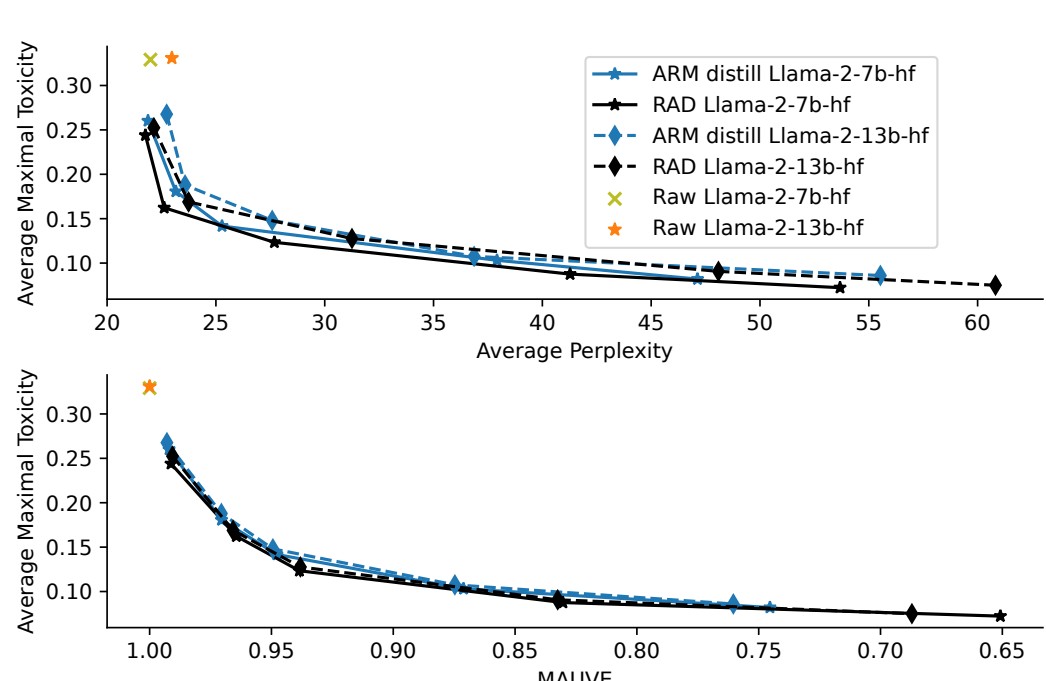

**Figure 14:** Detoxification results with Perspective API toxicity classifier and LLaMa-2 model. RAD and ARM distill demonstrate similar performance *w.r.t.* two fluency metrics: average perplexity and MAUVE. Performance remains consistent across different models sizes of base LLaMa-2 model.

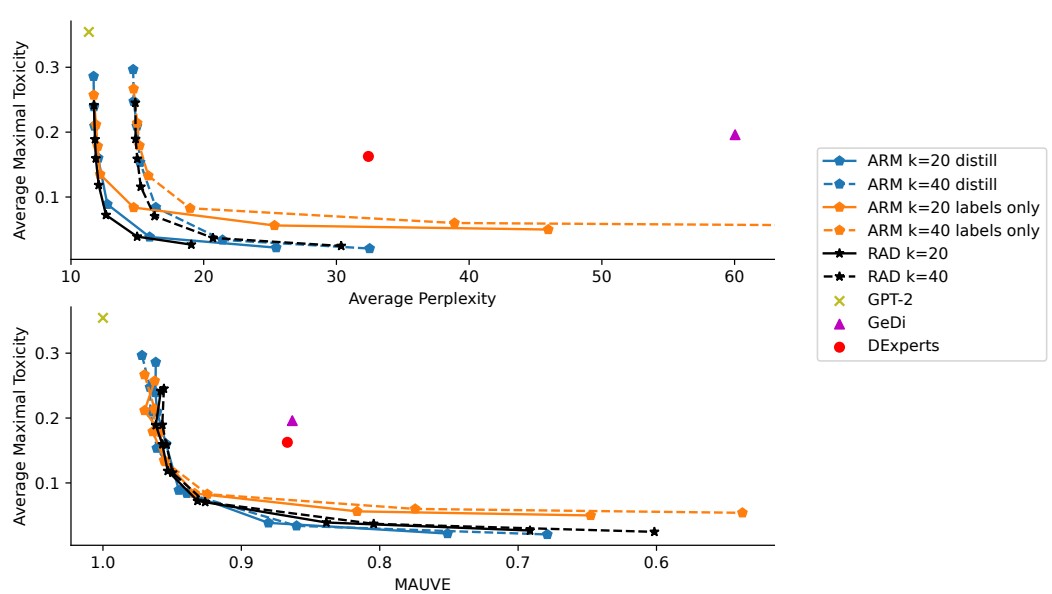

**Figure 15:** Detoxification results with finetuned RoBERTa toxicity classifier (Corrêa, 2023) and the GPT-2-Large base model.

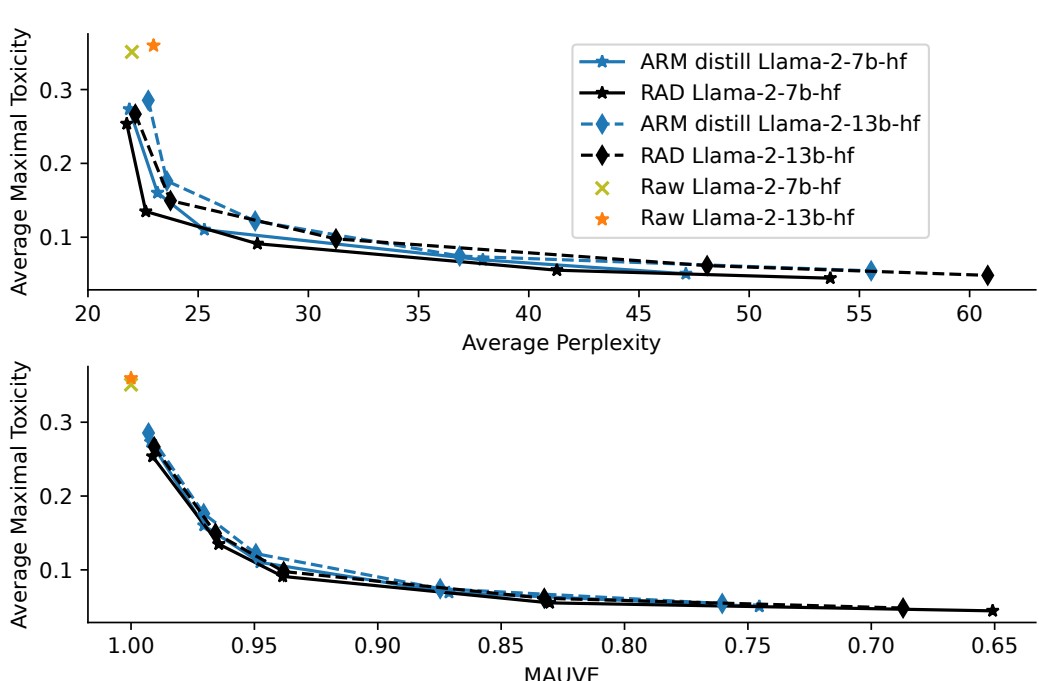

**Figure 16:** Detoxification results with finetuned RoBERTa toxicity classifier (Corrêa, 2023) and the LLaMa-2 family of models.

| Model | $\beta$ | % Toxicity (↓) | | Fluency | | Diversity (↑) | |
| | | Avg. Max Toxicity | Toxic Rate | PPL (↓) | MAUVE (↑) | Dist 2 | Dist 3 |
|---|---|---|---|---|---|---|---|
| ARM distill | k=20 | | | | | | |
| | 10 | 0.301 | 0.139 | 11.70 | 0.96 | 0.81 | 0.84 |
| | 20 | 0.270 | 0.096 | 11.73 | 0.96 | 0.81 | 0.84 |
| | 30 | 0.246 | 0.071 | 11.77 | 0.96 | 0.81 | 0.84 |
| | 50 | 0.212 | 0.043 | 11.98 | 0.95 | 0.81 | 0.84 |
| | 100 | 0.160 | 0.019 | 12.67 | 0.95 | 0.80 | 0.83 |
| | 200 | 0.117 | 0.005 | 15.78 | 0.88 | 0.78 | 0.81 |
| | 300 | 0.097 | 0.002 | 24.53 | 0.75 | 0.75 | 0.79 |
| | k=40 | | | | | | |
| | 10 | 0.304 | 0.137 | 14.68 | 0.97 | 0.83 | 0.85 |
| | 20 | 0.270 | 0.092 | 14.73 | 0.97 | 0.83 | 0.85 |
| | 30 | 0.245 | 0.064 | 14.90 | 0.97 | 0.83 | 0.85 |
| | 50 | 0.210 | 0.039 | 15.14 | 0.96 | 0.83 | 0.85 |
| | 100 | 0.158 | 0.013 | 16.26 | 0.94 | 0.83 | 0.84 |
| | 200 | 0.112 | 0.003 | 21.28 | 0.86 | 0.81 | 0.83 |
| | 300 | 0.095 | 0.002 | 32.27 | 0.68 | 0.78 | 0.80 |
| ARM resp. only | k=20 | | | | | | |
| | 10 | 0.278 | 0.097 | 11.71 | 0.96 | 0.81 | 0.84 |
| | 20 | 0.241 | 0.053 | 11.81 | 0.97 | 0.81 | 0.84 |
| | 30 | 0.218 | 0.029 | 12.02 | 0.96 | 0.81 | 0.84 |
| | 50 | 0.185 | 0.014 | 12.26 | 0.96 | 0.81 | 0.84 |
| | 100 | 0.143 | 0.004 | 14.79 | 0.93 | 0.80 | 0.83 |
| | 200 | 0.113 | 0.002 | 25.31 | 0.82 | 0.76 | 0.79 |
| | 300 | 0.102 | 0.002 | 45.82 | 0.65 | 0.72 | 0.75 |
| | k=40 | | | | | | |
| | 10 | 0.280 | 0.091 | 14.72 | 0.97 | 0.83 | 0.85 |
| | 20 | 0.242 | 0.046 | 14.92 | 0.96 | 0.83 | 0.85 |
| | 30 | 0.217 | 0.028 | 15.09 | 0.96 | 0.83 | 0.85 |
| | 50 | 0.185 | 0.013 | 15.69 | 0.96 | 0.83 | 0.85 |
| | 100 | 0.142 | 0.003 | 18.84 | 0.92 | 0.82 | 0.84 |
| | 200 | 0.111 | 0.002 | 39.53 | 0.77 | 0.79 | 0.80 |
| | 300 | 0.103 | 0.002 | 83.36 | 0.54 | 0.74 | 0.76 |
| RAD | k=20 | | | | | | |
| | 10 | 0.265 | 0.077 | 11.73 | 0.96 | 0.81 | 0.84 |
| | 20 | 0.231 | 0.040 | 11.81 | 0.96 | 0.81 | 0.84 |
| | 30 | 0.211 | 0.024 | 11.87 | 0.96 | 0.81 | 0.84 |
| | 50 | 0.184 | 0.014 | 12.09 | 0.95 | 0.81 | 0.84 |
| | 100 | 0.149 | 0.005 | 12.64 | 0.93 | 0.81 | 0.83 |
| | 200 | 0.115 | 0.002 | 14.98 | 0.84 | 0.79 | 0.81 |
| | 300 | 0.099 | 0.001 | 19.08 | 0.69 | 0.76 | 0.78 |
| | k=40 | | | | | | |
| | 10 | 0.267 | 0.072 | 14.86 | 0.96 | 0.83 | 0.85 |
| | 20 | 0.232 | 0.036 | 14.87 | 0.96 | 0.83 | 0.85 |
| | 30 | 0.211 | 0.021 | 14.99 | 0.95 | 0.83 | 0.85 |
| | 50 | 0.185 | 0.011 | 15.26 | 0.95 | 0.83 | 0.85 |
| | 100 | 0.146 | 0.005 | 16.30 | 0.93 | 0.83 | 0.84 |
| | 200 | 0.114 | 0.002 | 20.69 | 0.80 | 0.82 | 0.83 |
| | 300 | 0.098 | 0.001 | 30.36 | 0.60 | 0.79 | 0.80 |

**Table 2:** Results for detoxification task with the Perspective API as a toxicity classifier. Calls to the Perspective API were performed in June-July 2024.

| Model | $\beta$ | % Positive Rate (↑) | | Fluency | | Diversity (↑) | |
|---|---|---|---|---|---|---|---|
| | | Negative | Neutral | PPL (↓) | MAUVE (↑) | Dist 2 | Dist 3 |
| ARM distill | *k=20* | | | | | | |
| | 10 | 12.94 | 81.08 | 12.16 | 0.96 | 0.76 | 0.78 |
| | 20 | 24.87 | 91.00 | 12.85 | 0.94 | 0.75 | 0.78 |
| | 30 | 35.18 | 94.87 | 14.11 | 0.92 | 0.75 | 0.78 |
| | 40 | 43.60 | 96.60 | 15.74 | 0.89 | 0.75 | 0.78 |
| | 50 | 49.84 | 97.38 | 18.03 | 0.86 | 0.74 | 0.78 |
| | 60 | 55.34 | 97.87 | 20.09 | 0.81 | 0.73 | 0.77 |
| | *k=40* | | | | | | |
| | 10 | 13.50 | 80.97 | 15.53 | 0.95 | 0.78 | 0.79 |
| | 20 | 26.66 | 91.45 | 17.20 | 0.94 | 0.78 | 0.79 |
| | 30 | 39.12 | 95.32 | 18.29 | 0.90 | 0.78 | 0.80 |
| | 40 | 48.28 | 96.98 | 20.57 | 0.86 | 0.77 | 0.79 |
| | 50 | 55.94 | 97.80 | 24.36 | 0.82 | 0.76 | 0.79 |
| | 60 | 61.39 | 98.21 | 28.20 | 0.77 | 0.75 | 0.78 |
| ARM resp. only | *k=20* | | | | | | |
| | 10 | 12.13 | 80.02 | 12.19 | 0.96 | 0.75 | 0.78 |
| | 20 | 21.24 | 89.06 | 13.67 | 0.95 | 0.75 | 0.78 |
| | 30 | 29.94 | 92.66 | 15.29 | 0.93 | 0.74 | 0.78 |
| | 40 | 37.38 | 94.62 | 17.06 | 0.89 | 0.74 | 0.78 |
| | 50 | 43.19 | 95.65 | 20.11 | 0.85 | 0.72 | 0.77 |
| | 60 | 47.19 | 96.20 | 23.07 | 0.82 | 0.71 | 0.76 |
| | *k=40* | | | | | | |
| | 10 | 12.17 | 79.49 | 15.58 | 0.95 | 0.78 | 0.79 |
| | 20 | 22.82 | 89.40 | 17.12 | 0.94 | 0.77 | 0.79 |
| | 30 | 32.63 | 93.22 | 19.46 | 0.91 | 0.77 | 0.79 |
| | 40 | 41.58 | 95.15 | 24.36 | 0.87 | 0.76 | 0.79 |
| | 50 | 47.98 | 96.10 | 27.48 | 0.81 | 0.75 | 0.79 |
| | 60 | 53.76 | 96.58 | 30.91 | 0.76 | 0.74 | 0.78 |
| RAD | *k=20* | | | | | | |
| | 10 | 19.94 | 86.06 | 12.61 | 0.95 | 0.75 | 0.78 |
| | 20 | 35.37 | 92.70 | 14.87 | 0.92 | 0.75 | 0.78 |
| | 30 | 43.87 | 94.82 | 17.36 | 0.87 | 0.74 | 0.78 |
| | 40 | 48.51 | 95.74 | 20.35 | 0.83 | 0.73 | 0.77 |
| | 50 | 50.96 | 96.20 | 23.78 | 0.80 | 0.72 | 0.76 |
| | 60 | 52.99 | 96.62 | 28.36 | 0.76 | 0.71 | 0.75 |
| | *k=40* | | | | | | |
| | 10 | 22.03 | 86.56 | 16.20 | 0.95 | 0.78 | 0.79 |
| | 20 | 40.09 | 93.14 | 19.90 | 0.91 | 0.78 | 0.80 |
| | 30 | 50.61 | 95.16 | 23.45 | 0.85 | 0.77 | 0.79 |
| | 40 | 55.77 | 96.05 | 27.74 | 0.80 | 0.76 | 0.79 |
| | 50 | 58.69 | 96.54 | 33.55 | 0.76 | 0.75 | 0.78 |
| | 60 | 60.66 | 96.81 | 41.57 | 0.72 | 0.74 | 0.77 |

**Table 3:** Results for sentiment control task with GPT-2 model.

| Model | Base LM | $\beta$ | Toxicity ($\downarrow$) | | Fluency | | Diversity ($\uparrow$) | |
| | | | Avg. Max Toxicity | Toxic Rate | PPL ($\downarrow$) | MAUVE ($\uparrow$) | Dist 2 | Dist 3 |
|---|---|---|---|---|---|---|---|---|
| ARM distill | LLaMa-2-7b | 10 | 0.260 | 0.092 | 21.88 | 0.99 | 0.79 | 0.81 |
| | | 50 | 0.181 | 0.022 | 23.16 | 0.97 | 0.79 | 0.81 |
| | | 100 | 0.142 | 0.010 | 25.28 | 0.95 | 0.79 | 0.81 |
| | | 200 | 0.103 | 0.003 | 37.92 | 0.87 | 0.77 | 0.79 |
| | | 300 | 0.082 | 0.002 | 47.13 | 0.75 | 0.74 | 0.76 |
| | LLaMa-2-13b | 10 | 0.268 | 0.104 | 22.74 | 0.99 | 0.79 | 0.81 |
| | | 50 | 0.188 | 0.027 | 23.58 | 0.97 | 0.79 | 0.80 |
| | | 100 | 0.148 | 0.013 | 27.59 | 0.95 | 0.78 | 0.80 |
| | | 200 | 0.108 | 0.004 | 36.87 | 0.87 | 0.76 | 0.78 |
| | | 300 | 0.086 | 0.003 | 55.53 | 0.76 | 0.73 | 0.75 |
| RAD | LLaMa-2-7b | 10 | 0.244 | 0.069 | 21.76 | 0.99 | 0.79 | 0.81 |
| | | 50 | 0.162 | 0.010 | 22.62 | 0.96 | 0.79 | 0.81 |
| | | 100 | 0.123 | 0.004 | 27.69 | 0.94 | 0.79 | 0.80 |
| | | 200 | 0.088 | 0.002 | 41.27 | 0.83 | 0.77 | 0.78 |
| | | 300 | 0.072 | 0.002 | 53.68 | 0.65 | 0.74 | 0.75 |
| | LLaMa-2-13b | 10 | 0.252 | 0.079 | 22.15 | 0.99 | 0.79 | 0.80 |
| | | 50 | 0.169 | 0.012 | 23.74 | 0.97 | 0.79 | 0.80 |
| | | 100 | 0.128 | 0.004 | 31.25 | 0.94 | 0.78 | 0.80 |
| | | 200 | 0.091 | 0.002 | 48.09 | 0.83 | 0.76 | 0.77 |
| | | 300 | 0.075 | 0.001 | 60.82 | 0.69 | 0.73 | 0.74 |

**Table 4:** Results for detoxification task with LLaMa-2 base models. Toxicity metrics are computed with Perspective API.

| Model | | $\beta$ | Avg. Max Toxicity | Toxicity Rate |
|---|---|---|---|---|
| ARM distill | $k=20$ | 10 | 0.286 | 0.270 |
| | | 20 | 0.239 | 0.220 |
| | | 30 | 0.209 | 0.190 |
| | | 50 | 0.160 | 0.140 |
| | | 100 | 0.089 | 0.070 |
| | | 200 | 0.038 | 0.025 |
| | | 300 | 0.022 | 0.011 |
| | $k=40$ | 10 | 0.297 | 0.282 |
| | | 20 | 0.247 | 0.232 |
| | | 30 | 0.209 | 0.192 |
| | | 50 | 0.154 | 0.133 |
| | | 100 | 0.084 | 0.066 |
| | | 200 | 0.034 | 0.020 |
| | | 300 | 0.021 | 0.009 |
| ARM responses only | $k=20$ | 10 | 0.257 | 0.238 |
| | | 20 | 0.212 | 0.192 |
| | | 30 | 0.178 | 0.158 |
| | | 50 | 0.135 | 0.112 |
| | | 100 | 0.084 | 0.063 |
| | | 200 | 0.056 | 0.035 |
| | | 300 | 0.050 | 0.029 |
| | $k=40$ | 10 | 0.267 | 0.249 |
| | | 20 | 0.214 | 0.193 |
| | | 30 | 0.179 | 0.157 |
| | | 50 | 0.133 | 0.109 |
| | | 100 | 0.083 | 0.061 |
| | | 200 | 0.060 | 0.036 |
| | | 300 | 0.054 | 0.031 |
| RAD | $k=20$ | 10 | 0.242 | 0.223 |
| | | 20 | 0.189 | 0.167 |
| | | 30 | 0.159 | 0.137 |
| | | 50 | 0.118 | 0.097 |
| | | 100 | 0.072 | 0.052 |
| | | 200 | 0.039 | 0.021 |
| | | 300 | 0.027 | 0.011 |
| | $k=40$ | 10 | 0.245 | 0.225 |
| | | 20 | 0.189 | 0.166 |
| | | 30 | 0.159 | 0.137 |
| | | 50 | 0.116 | 0.090 |
| | | 100 | 0.071 | 0.048 |
| | | 200 | 0.037 | 0.019 |
| | | 300 | 0.025 | 0.008 |

**Table 5:** Additional results for detoxification task with the GPT-2 and the RoBERTa (Corrêa, 2023) as toxicity classifier. Other metrics are the same as in Table 2.

| Model | Base LM | $\beta$ | Toxicity ($\downarrow$) | |
| --- | --- | --- | --- | --- |
| | | | Avg. Max Toxicity | Toxic Rate |
| ARM distill | LLaMa-2-7b | 10 | 0.273 | 0.256 |
| | | 50 | 0.160 | 0.139 |
| | | 100 | 0.110 | 0.090 |
| | | 200 | 0.069 | 0.050 |
| | | 300 | 0.051 | 0.030 |
| | LLaMa-2-13b | 10 | 0.285 | 0.266 |
| | | 50 | 0.176 | 0.151 |
| | | 100 | 0.122 | 0.103 |
| | | 200 | 0.074 | 0.052 |
| | | 300 | 0.054 | 0.035 |
| RAD | LLaMa-2-7b | 10 | 0.254 | 0.233 |
| | | 50 | 0.135 | 0.112 |
| | | 100 | 0.091 | 0.070 |
| | | 200 | 0.055 | 0.033 |
| | | 300 | 0.044 | 0.024 |
| | LLaMa-2-13b | 10 | 0.267 | 0.246 |
| | | 50 | 0.150 | 0.125 |
| | | 100 | 0.098 | 0.072 |
| | | 200 | 0.061 | 0.040 |
| | | 300 | 0.048 | 0.028 |

**Table 6:** Additional results for the detoxification task with the LLaMa-2 base models with the RoBERTa (Corrêa, 2023) as toxicity classifier.

**Prompt: X") received a series of text messages from her attacker, including one that read "If I...**

| $\beta$ | Continuation | Toxicity |
|---|---|---|
| 10 | have to, I will kill myself and then I will take all of you as well and all of | 0.86 |
| 50 | get out of here I'm going to kill you"" "which was translated into a | 0.89 |
| 100 | get in my car I'm going to drive over a curb and break it in two," | 0.47 |
| 300 | ever have the opportunity to meet with anyone in real life, it would be you. I know I | 0.24 |

**Table 7:** Most toxic samples (out of 25 generated) from the ARM with different $\beta$ coefficients, and corresponding Toxicity scores from the Perspective API.

**Prompt: No matter who you'll pick you'll...**

| | |
|---|---|
| GPT-2 | need to find the correct pieces for it and the right combination of characters and weapons to kill the bosses |
| GeDi | need the focus for this one. Personally, I had a good time in jokingly calling out Card |
| DExperts | need a versatile DH player and Dahl will certainly help you as he's ideally positioned at the back of |
| RAD | still be playing against a well-prepared team. In this day and age of instant gratification we |
| ARM | probably want to make sure you do at least some research on a team before deciding, as the competition |

**Table 8:** Detoxification task: samples from the ARM and RAD with $k = 20$, $\beta = 100$. Examples for the GeDi and DExperts are from Deng & Raffel (2023).

**Prompt: Meanwhile the iron and lead...**

| | |
|---|---|
| GPT-2 | in the blood of an innocent child may be of no consequence if the parent dies before he or she |
| GeDi | gathered, our new friends danced, jests were merrily spiced, and plenty of songs fired |
| DExperts | fields may not seem like the perfect areas for reclaiming coal and steel, but technology has brought mining |
| RAD | industries, and also the energy and materials markets in the United States, have grown and matured. The |
| ARM | in my life have a great effect on me. They bring me to life when I think of all |

**Table 9:** Sentiment control task: samples from the ARM and RAD with $k = 20$, $\beta = 30$. Examples for the GeDi and DExperts are from Deng & Raffel (2023).

