# OpenReview forum: "Efficient Controlled Language Generation with Low-Rank Autoregressive Reward Models"
_ICLR.cc/2025/Conference — Submitted to ICLR 2025_

### Official Review · Reviewer_24mX · 2024-10-30

**Soundness:** 4
**Presentation:** 4
**Contribution:** 3
**Rating:** 8
**Confidence:** 4

**Summary:**

* This paper proposes ARM (Autoregressive Reward Model), a low-rank approximation of RAD (Reward Augmented Decoding) for reward-guided controlled decoding.
    * ARM is motivated by casting the reward modeling problem into matrix completion and the empirical insight that the “reward matrix” to complete is low-rank.
    * ARM has a more efficient inference complexity of $O(L)$ in terms guiding a length-$L$ generation and considering $k$ next-token candidates at each step. In comparison, RAD has a complexity of $O(Lk)$.
    * ARM is trained with two steps: (1) train with RAD’s objective to estimate next-token reward given the prefix and (2) distill from a RAD teacher.
* They conduct two sets of controlled generation experiments: Detoxification as evaluated on the RealToxicityPrompt dataset and Sentiment Control as evaluated on the OpenWebText’s prompts. They evaluate the effectiveness of ARM controlled generation with the fluency-controlled attribute (toxicity/sentiment) tradeoff in comparison with RAD and previous methods such as GeDi and DExperts.
* Their experimetnal results suggest that ARM achieves similar if not better fluency-controlled attribuite tradeoff compared to RAD and previous methods on these two tasks. They also show that the resultant reward matrix from ARM is indeed low rank as designed.

**Strengths:**

* The paper is clearly written and well-presented.
* The experiments are extensive with sensible setups and evaluation procedure.
* The proposed method ARM performs on par with the more expensive RAD alternative at much better inference complexity.

**Weaknesses:**

* It would be good to have a expanded discussion of some experimental results (See Questions below).
* It would be good to include the compute required for training/evaluating these models.

**Questions:**

1. Are there hyper-parameters similar to the beta in RAD & ARM for GeDi and DExperts to trade-off fluency for detoxicity? If so, it would be good to show their operating points in Figure 3 to make a stronger case.
2. In Figure 3, increasing k seems to induce a right-ward shift (i.e., higher perplexity at similar toxiticy level). Could you help me understand why?
3. Line 431: “we observe that regularization effectively decreases the rank of R_ARM which might explain the higher fluency of regularized model.” This suggests lower-rank approximation is better in terms of fluency. Can you expand on this point and help me understand why this is the case?
4. What is the compute time required for running evaluation and training ARM?

---

> ### Author Response · Authors · 2024-11-20
> **Response**
>
> We thank the reviewer for their feedback. We appreciate that they highlight the clarity of our paper, as well as the extensiveness of our experiments and evaluation.
>
> > Are there hyper-parameters similar to the beta in RAD & ARM for GeDi and DExperts to trade-off fluency for detoxicity?  If so, it would be good to show their operating points in Figure 3 to make a stronger case.
>
> We agree that it will make the Figure 3 better if we include the trade-off plot for the DExperts model (which is known to outperform GeDi). To give a quicker response, we add a preliminary figure (see Figure 8 in Appendix) by running the evaluation over 1000 prompts for ARM vs DExperts. We observe that DExperts start to diverge early from ARM resulting in lower effectiveness of the language modeling-based guidance. We aim to include the DExperts evaluated over all toxicity prompts for the final version.
>
> > in Figure 3, increasing k seems to induce a right-ward shift (i.e., higher perplexity at similar toxiticy level). Could you help me understand why?
>
> This is because for larger k there are less probable tokens that can be produced by the model. The relative performance of ARM compared to RAD remains similar for both settings. For simplicity of the Figure 3, we only leave k=20 case, and move k=40 case to the Appendix, as was suggested by another reviewer.
>
> > Line 431: “we observe that regularization effectively decreases the rank of R_ARM which might explain the higher fluency of regularized model.” This suggests lower-rank approximation is better in terms of fluency. Can you expand on this point and help me understand why this is the case?
>
> We meant to highlight that our regularization is aimed to push the model to abstain more by regularizing the prediction towards rank-1 output (predicting the baseline score for every next token, which does not modify the base model distribution). We will clarify: “Particularly, a very strong regularization would result in the model always predicting the baseline score for each of the next tokens (corresponds to the rank-1 output), which does not modify the original distribution of the model (the best fluency).”
>
> > What is the compute time required for running evaluation and training ARM?
>
> Here, we report the training/generation time for the detoxification task. For generation, we report the time to obtain a single point on the trade-off plot (a single value of $\beta$).
>
> Training: for ARM and RAD similar and approx. 12 hours per epoch on 1 GPU.
>
> Generation speed for ARM vs RAD:
>
> | Model Combination                     | k   | Decoding Method | Time    | Relative Duration to ARM |
> |---------------------------------------|-----|-----------------|---------|--------------------------|
> | **GPT Large + GPT Small Expert** | 20,40  | ARM             | ~1h 35m  | 1.0x                     |
> |                                       | 20  | RAD             | 8h 22m  | ~5.3x                    |
> |                                       | 40  | RAD             | 14h 36m | ~9.2x                    |
> | **LLaMa-2 13b + TinyLLaMa Expert**    | 40  | ARM             | 4h 23m  | 1.0x                     |
> |                                       | 40  | RAD             | 41h 14m | ~9.4x                    |

---

### Official Review · Reviewer_C7CW · 2024-10-31

**Soundness:** 3
**Presentation:** 3
**Contribution:** 2
**Rating:** 3
**Confidence:** 4

**Summary:**

The paper explores an efficient way to guide language models using low-rank reward models for controlled text generation. Traditional approaches like Reward Augmented Decoding (RAD) is computationally expensive because it processes each candidate token individually (each token requires a forward pass using the reward model). The authors propose an alternative method, the low-rank Autoregressive Reward Model (ARM), which simplifies the process by representing the reward model with fewer parameters. This change maintains performance in tasks like detoxification and sentiment control while significantly speeding up decoding. ARM is shown to match RAD in quality but requires fewer computational resources.

**Strengths:**

1. The paper is well-written and easy to follow. The authors conduct thorough experiments to demonstrate the effectiveness and efficiency of the proposed approach. Enough experiential details are provided for others to replicate the results.

**Weaknesses:**

1. The proposed approach offers limited novelty. The authors suggest enhancing the prediction efficiency of the reward model by scoring all potential next tokens in a single forward pass through the language model's backbone (Section 3.2). However, this concept has already been explored in prior research [1, 2], which treats the prefix score as an action-value function using a language model as the backbone.

- Systematic Rectification of Language Models via Dead-end Analysis, ICLR 2023
- Controlled Decoding from Language Models, ICML 2024

2. Efficiency improvement seems limited as most of the computational cost comes from the LM backbone.

**Questions:**

In Section 3.3 on ARM training, is the language model backbone also fine-tuned, or is it kept frozen?

---

> ### Author Response · Authors · 2024-11-20
> **Response**
>
> We thank the reviewer for their feedback. We are happy to hear that our work is easy to follow; that we conduct thorough experiments and provide enough experiential details.
>
> > In Section 3.3 on ARM training, is the language model backbone also fine-tuned, or is it kept frozen?
>
> We finetune all parameters of the reward model except input/output embeddings, as we state in section 4.1.
>
> > The proposed approach offers limited novelty.
>
> We appreciate a valuable connection to the reinforcement learning-based controlled generation. While their contribution is not the parametrization of Q_D, Cao et al. 2023 use a language model architecture similar to ours, DExperts, GeDi. Our work zooms in on the parametrization choice, we explore the implications of rank, and we find a surprising result in terms of the rank of reward models. This is what we bring as truly novel, in contrast to simply predicting the scores in one go (e.g. DExperts, GeDi already did this).
> We discuss the relationship to this work in the revision. We also add the work of Mudgal et al. 2024, where they use prefix scorers in a way similar to how less efficient RAD or FUDGE are used. We hope our work will inform the reinforcement learning branch of controlled generation.
>
> - Cao et al. 2023. Systematic Rectification of Language Models via Dead-end Analysis, ICLR 2023
> - Mudgal et al. 2024. Controlled Decoding from Language Models, ICML 2024
>
> > Efficiency improvement seems limited as most of the computational cost comes from the LM backbone
> In our experiments, we observe a high difference between ARM and RAD in terms of evaluation speed.
>
> For generation, here we report the time to obtain a single point on the trade-off plot (a single value of $\beta$).
>
> Decoding for ARM vs RAD:
>
> | Model Combination                     | k   | Decoding Method | Time    | Relative Duration to ARM |
> |---------------------------------------|-----|-----------------|---------|--------------------------|
> | **GPT Large + GPT Small Expert** | 20,40  | ARM             | 1h 35m  | 1.0x                     |
> |                                       | 20  | RAD             | 8h 22m  | ~5.3x                    |
> |                                       | 40  | RAD             | 14h 36m | ~9.2x                    |
> | **LLaMa-2 13b + TinyLLaMa Expert**    | 40  | ARM             | 4h 23m  | 1.0x                     |
> |                                       | 40  | RAD             | 41h 14m | ~9.4x                    |

---

> > ### Comment · Reviewer_C7CW · 2024-11-26
> >
> > I appreciate the authors' response. While the reward model analysis is interesting, I do not believe it qualifies as a major contribution. Regarding the efficiency improvement, RAM employs an output head on top of a language model to compute scores for all next-token candidates in a single forward pass. However, this idea/implementation is not novel, as it has already been explored in previous work (see weaknesses).
> >
> > I will maintain my current score.

---

### Official Review · Reviewer_xYFT · 2024-11-03

**Soundness:** 2
**Presentation:** 2
**Contribution:** 2
**Rating:** 5
**Confidence:** 3

**Summary:**

This paper considers the problem of steering a trained language model to generate better outputs. At a high level, this paper does this by proposing an improvement to the reward-augmented decoding (RAD) paradigm of Deng and Raffel (2023). Specifically, they propose a "low-rank autoregressive reward model" (ARM) for guided decoding. In a sentence, the ARM models reward scores for next token candidates. They provide theoretical support for proposing a lower-rank version of RAD, and they investigate two ways of doing this in practice:
1. They propose distilling RAD into ARM
2. They propose training ARM from scratch

**Strengths:**

In my opinion: at a high level, the strengths of the paper included the motivation, the clarity of the writing, and the clarity of the mathematical formulations. In more detail--

1. I thought that the introduction did a great job setting the scene and motivating the research direction. For example, the paper says: _"Control over LLMs can be roughly divided into methods which modify the original model via finetuning, and decoding-time solutions, which do not modify the parameters of the original model."_ These types of high-level summaries/divisions of "where the field is at" are very useful for providing context, and the paper is full of good things like this.

2. I thought that the motivation here as very high quality as well: _"the RAD approach is flexible enough to represent a large space of reward matrices including those of high rank. However, when we empirically measure the rank of the reward matrix learned by RAD, it appears to be low-rank."_ This was a very clear way of highlighting, early on in the paper, the high level motivation of the "main big idea" in the paper--doing a lower-rank version of RAD.

3. The formulation of _"Section 2.1: Guided Decoding with External Experts"_ was very high quality. Specifically, the formulas were all explained and motivated very well. It was delightful to read a math section as clear as this one.

4. Table 1 and Figure 6 are clear, and make the author's point about increased efficiency with the ARM method very clearly.

**Weaknesses:**

In my opinion: at a high level, the weaknesses of this paper were that rigorous theoretical justification for considering a low-rank version of RAD were deferred to the appendix and not adequately summarized in the main body of the paper; the paper proposes two different ways of doing low-rank RAD (distilling, and training from scratch) and this should be more prominently stated (e.g. in a "Our main contributions are as follows..." bulleted list); and the graphs in the paper are so difficult to understand that it is unclear to me to what extent the experimental results show any performance improvement.

In more detail--

1. _Section 3.1: Analysis of RAD_ could do a better job providing rigorous justification for why low-rank learning is more theoretically sound than what is considered in the original RAD paper. Delegating most of the rigorous mathematics to the appendix is problematic in this case, because that's where almost all of the rigorous claims where, if I understand correctly. I understand wanting to save space, but the paper has almost an entire page worth of more space... I think that it would seriously strengthen the paper to at least include statements of the "main theorems/propositions" and then delegate the proofs to the appendix. Also, framing this section around theorems/propositions would break it up and make the logical flow easier to follow. For example, this section includes the following phrases: _"Particularly, the incompleteness of PΩ(R) makes it easier for a reward model to learn a low rank approximation, especially for unique prefixes x, as we demonstrate in Appendix B.1. To better understand this phenomenon, we would like to understand whether the PΩ(R) can be fit with the low-rank model. In Appendix B.2, we demonstrate that indeed incomplete PΩ(R) matrix can be fit with the low-rank matrix factorization with a small error. This implies that the training dataset can be fit by a model that produces low-rank \hat R , regardless of the specifics of said model."_ To make the arguments rigorous and convincing, I think all these claims should all be spelled out in precise detail in the main body of the paper (using rigorous statements, instead of vague sentences like those I quoted), and then defer all proofs to the appendix. This is standard practice.

2. In Equation (8), I don't think RAD teacher \tilde r(x) is defined, and it's very confusing what it is. It wasn't until I got to the experiments section that I realized that the paper considers two paradigms--1) distilling RAD into "low-rank RAD" (i.e. ARM), and 2) training a low-rank RAD (i.e. an ARM) from scratch. I think this should be explicitly stated somewhere in that section that (7) and (8) are completely different training paradigms.

3. (The main weakness, in my opinion) the graphs/results are very unclear. In Figure 3, the figure is too busy for me to understand what's going on, and I've been spent considerable time trying. I recommend having 1 set of graphs with k=20, and a second set with k=40 (and putting set in the appendix) and zooming in so that we can actually see the order of magnitude differences between the curves that are being compared in the graph. I'm not sure what point this graph tries to convey; the paper needs to explain what each curve's trend represents, relative to the other curve's trends. And it is unclear to me how each of the individual points on the curves is obtained (I know there is some third, unseen parameter which parameterizes the curves and controls the toxicity/perplexity and toxicity/MAUVE tradeoffs. Why is this third parameter not explicitly mentioned in the figures? Further, is it significant that this third parameter doesn't let ARM distilled version's perplexity go past the mid-30's?) These graphs raise more questions than answers, and I don't understand how they provide experimental evidence that ARM is better than RAD. If the authors can clarify that point, I'd appreciate it. But without clarity on these experimental results, I don't have any confidence that performance of the ARM version is on par with the RAD version; my only take-away from these graphs are that every model follows an approximately 1/x shaped tradeoff curve, but that's alone is not sufficient to demonstrate the relative success of the method.

Some minor points:

1. On line 042, it says _"reward augmented generation (RAD)"_ when I think it should say "reward augmented decoding (RAD)".

2. In the next version of the draft, the paper should have every single equation numbered. E.g. I want to be able to reference the definition for D_f using its equation (see my questions), but I can't. This will let other researchers discuss the paper's contents more precisely.

3. If the authors want this paper to be self-contained, perhaps they can add a concrete example, with a given small prefix, to illustrate what goes on in Section 2.1: RAD Training. As someone who was not familiar with RAD until reading this paper, I had to read through this section a few times to understand what was going on, it is a tad dense.

**Questions:**

1. Re: the Definition of D_f, can the authors please provide for me some motivation for why we want to use the same reward y regardless of how long the prefix x is? It seems like this might add spurious correlations to the training data. (I don't think this necessarily needs to go in the paper, this is more of a question I'm asking to gain a deeper understanding of how RAD works... this seems a major shortcoming of the entire RAD pipeline, do you know if researchers have considered alternative ways of assigning numerical rewards to the shortened prefixes?)

2. Are equations (4), (5), and (6) all equal? As in, are they just versions of the same exact same equations? (It appears to me that they are...) If indeed they are, this should be noted somewhere in the paper.

3. In Appendix C.4, what is $\sigma_1$?

4. Re: Appendix C--can the authors please explain in more detail what issues might arise when estimating the numerical rank? (i.e. what types of things can go wrong. And how did the authors confirm that these things did not go wrong when they were estimating the numerical ranks for their experiments.)

---

> ### Author Response · Authors · 2024-11-20
> **Response (1/2)**
>
> We would like to thank the reviewer for the feedback. We are happy to hear that they find our motivation, writing, and mathematical formulations to be clear!
>
> > Delegating most of the rigorous mathematics to the appendix is problematic. I think that it would seriously strengthen the paper to at least include statements of the "main theorems/propositions" and then delegate the proofs to the appendix.
>
> We thank the author for this suggestion! We followed your advice and moved some statements from the Appendix to the main text and we believe this indeed strengthens our work. We refer reviewer to an updated version of the pdf (3.1 Analysis of RAD).
>
> > In Equation (8), I don't think RAD teacher \tilde r(x) is defined. I think this should be explicitly stated somewhere in that section that (7) and (8) are completely different training paradigms.
>
> We apologize for the missing introduction of the distillation task, we improve this in section 3.3 (ARM Training).
>
> > (The main weakness, in my opinion) the graphs/results are very unclear. In Figure 3, the figure is too busy for me to understand what's going on.  I recommend having 1 set of graphs with k=20, and a second set with k=40 (and putting set in the appendix) and zooming in so that we can actually see the order of magnitude differences between the curves that are being compared in the graph.
>
> This is a great suggestion, we refactored the plots and moved parts of the plot with k=40 to the Appendix.
>
> >  it is unclear to me how each of the individual points on the curves is obtained (I know there is some third, unseen parameter which parameterizes the curves)
>
> You are right, there is a third scalar parameter, which defines a position on the trade-off plot. We didn’t include these parameters in the main figures to not make a plot more complicated, and also because the ranges of these control coefficients are not directly comparable between models (one compares the two trade-off lines as a whole: fixes one metric and compares another one). We included the used trade-off parameters beta in the Appendix (Tables 2-6).
>
> > I'm not sure what point this graph tries to convey; the paper needs to explain what each curve's trend represents, relative to the other curve's trends.
>
> Thank you for this comment, we add an introductory paragraph in Section 4.4. Results to explain how to read the plots. These kind of trade-off plots are standard to compare models in control generation literature, e.g. see Figure 2 in RAD, or Figure 5 in DExperts. In our work, to compare RAD with ARM, we take several beta coefficients and plot the trade-off lines. Then, we show that two lines are close to each other: there is a little gap between the two plots in most of the figures, which allows us to empirically conclude that ARM indeed well approximates RAD.
>
> > Further, is it significant that this third parameter doesn't let ARM distilled version's perplexity go past the mid-30's?)
>
> If we continue to increase the control coefficient $\beta$, we expect the perplexity of ARM to continue to grow (the same for RAD).
>
> > Definition of D_f: can the authors please provide for me some motivation for why we want to use the same reward y regardless of how long the prefix x is?
>
> We agree that the setup of RAD might be not optimal and future work might improve on the particular choice of using the data. Nevertheless, we analyze the training objective of RAD and observe that in order to obtain optimal loss for the training dataset, the model would learn to predict the expected future response (Equation 5). We hope this provides more intuition to the RAD approach.
>
> > the paper should have every single equation numbered
>
> Thank you for this suggestion, we added numbering for all the equations.
>
> > Are equations (4), (5), and (6) all equal? As in, are they just versions of the same exact same equations?
>
> The equations you referred to are indeed versions of the same equation, we clarify this in the revision.
>
> > If the authors want this paper to be self-contained, perhaps they can add a concrete example, with a given small prefix, to illustrate what goes on in Section 2.1: RAD Training
>
> Thank you for this suggestion. We will add a figure with an example for the final version.

---

> > ### Author Response · Authors · 2024-11-20
> > **Response (2/2)**
> >
> > > In Appendix C.4, what is σ1 ?
> >
> > $\sigma_1$ is the largest singular value of the singular value decomposition, we clarify this, thank you.
> >
> > > Re: Appendix C--can the authors please explain in more detail what issues might arise when estimating the numerical rank?
> > (i.e. what types of things can go wrong. And how did the authors confirm that these things did not go wrong when they were estimating the numerical ranks for their experiments.)
> >
> > One issue is that the singular values, especially for the matrices coming from 32bit float precision neural network, will not be exactly zero, so this is why libraries like Numpy or PyTorch use a precision-based cutoff for singular values that should be considered indistinguishable from zero; we use the default such parameters. The other issue is that the number of rows in the reward matrices is very high and we follow the work of Finlayson et al. 2024 and estimate rank by sampling rows from the matrix. Different submatrices can have different ranks, but we sample i.i.d. to prevent this.
> > We clarify a bit potential pitfalls we could come up with, but we are not sure if this is what you meant, could you please clarify?
> >
> > We thank the author for valuable suggestions on how to improve the readability of the paper. We hope that our response and modifications to the paper address your concerns.
> >
> > - Finlayson et al. 2024. Closing the curious case of neural text degeneration. ICLR, 2024.

---

> ### Comment · Reviewer_xYFT · 2024-11-25
> **Response to Authors**
>
> Thank you for the detailed reply, and for updating the draft. Also, thanks for adding the clarifications I mentioned (e.g. the changes made to Section 3.1, Analysis of RAD--I think this makes the mathematical motivation clearer; also, specifying that there are two different training paradigms considered in the paper, distilling RAD into "low-rank RAD" (i.e. ARM), versus training a low-rank RAD (i.e. an ARM) from scratch). And I appreciate the explanation about estimating the rank, I'm satisfied there.
>
> Further questions about the experimental results:
> 1. I appreciate the refactoring of the graphs, they're visually simpler to parse now, thanks. However, Figure 4 is never referenced in the main text and so it isn't immediately clear how it supports the overall story, can you please add a reference to it somewhere in the PDF to explain how it fits into the overall story? (This is already done well with Figure 3)
> 2. Can you please help me understand how the trade-off plots (e.g. Figures 3 and 4) indeed show positive results? For example, take Figure 4, the second plot, for Average Perplexity versus Negative Prompt Positive Rate. I agree that visually there is little gap between RAD (the baseline) and ARM Distill and ARM resp. only (i.e. the train from scratch model). **However, if you look closer at the visual distance (i.e. the actual numerical differences) between the black RAD and the orange/blue curves, it looks like the distance can be as high as $0.05$, which corresponds to an approximately $0.05/0.4 = 12.5\\%$ performance difference.** I guess my skepticism/misunderstanding is: suppose that we don't have any sort of heuristics that say that "this high rank reward matrix is essentially just a low rank matrix." Then what's stopping us from distilling the high rank matrix into a low-rank matrix and losing 12.5% performance? I.e. how is the rank analysis actually useful, if it leads to any performance gap at all, let alone a 12.5% performance decrease, which seems very undesirable? Unless if there is some application where the speed up from the paper's method is sufficiently valuable as to offset a 12.5% performance decrease, but that's not my understanding here. (Alternatively, please let me know if I'm reading these graphs incorrectly.)
>
> Thanks again for answering my questions!

---

> > ### Author Response · Authors · 2024-11-26
> > **Response to reviewer**
> >
> > We thank the reviewer for continuing the discussion.
> >
> > >  Figure 4 is never referenced in the main text
> >
> > Thank you for attentive look! We fixed the reference to Figure 4 (line 468. sentiment control task results discussion), which got broken during revision.
> >
> > > Can you please help me understand how the trade-off plots (e.g. Figures 3 and 4) indeed show positive results?
> >
> > Our analysis suggests that ARM parametrization is expressive enough to match RAD, but it does not say that two models will be equivalent (there are multiple reward matrices compatible with the observed values of training reward matrix). As a positive result, we observe that our distilled ARM either match RAD or even demonstrate slightly higher quality (Figures 3,4: blue line) compared to RAD, which empirically demonstrates that the ARM parametrization is indeed capable of matching the quality of RAD while being more efficient.
> >
> > Indeed as we highlight on line 474, that ARM trained on original utterances underperforms compared to both distilled ARM model and sometimes RAD. As we write in response to reviewer BihH, one clear difference is that when training from data, we will see short contexts multiple times with different reward responses and must implicitly converge to their average, while in distillation, the teacher already performs this compression and provides a single deterministic target $\hat{r}(v|x)$ for every context $(x,v)$. We conjecture that this may lead to better-trained distilled models. It is possible that extra tuning or regularization would improve ARM trained on original responses. We added this discussion at the end of the Section 5.4.

---

> ### Comment · Reviewer_xYFT · 2024-12-01
> **Response to authors**
>
> I would like to thank the authors for answering my questions. My primary concern in my original review was the clarity of the graphs, and that has been addressed. Accordingly, I have raised raised my soundness score (1-->2).
>
> However, now that I have been able to more thoroughly examine the experimental results, I am more convinced that distillation is very valuable, but I am not entirely convinced that training ARM from scratch is valuable. In the abstract, the authors claim that *"our low-rank reward model performs on par with the more flexible RAD parametrization"*. But as I noted earlier, training ARM from scratch tends to underperform RAD significantly (approximately 12% on the given metrics). In contrast, distilling ARM from RAD tends to perform better than RAD, and I consider this a valuable result.
>
> I understand that the authors can't upload a revised paper version at this point in the rebuttal period. However, would the authors be willing to do one of the following:
> 1. Dampen the claims, especially in the abstract (e.g. changing the wording to the following-- *"distilling a low-rank reward model from the more flexible RAD parametrization performs on par with the teacher RAD parametrization"*, or something better written than that which conveys that it's not training from scratch that wins, but rather the distillation version)
> 2. OR--convince me that I am mistaken, and the training from scratch version of ARM indeed gives results on par with the more flexible RAD parametrization.
>
> Thanks!

---

> > ### Author Response · Authors · 2024-12-02
> > **Thanks for your feedback**
> >
> > We thank the reviewer for their feedback. We will correct our claims about empirical results and explicitly write that we observe that *distilling* a low-rank reward model from the more flexible RAD parameterization performs on par or better compared to the teacher RAD parameterization, while ARM trained on original labels can underperform compared to RAD.

---

> ### Comment · Reviewer_xYFT · 2024-12-01
> **Response to authors--why not a higher/lower score?**
>
> Additionally, the AC has indicated that they would like answers to these questions, I will provide them here:
>
> 1. *Why not a lower score?*--The ARM distillation results are compelling because they obtain better results than the original RAD parameterization, in addition to inference runtime speed up. As a result, these results would be valuable to the community.
> 2. *Why not a higher score?*--From the experimental results, it seems like training ARM from scratch can actually degrade downstream performance, significantly. These experimental results seem to contradict a central theoretical claim of the paper, that all the extra rank capacity is not needed in order to obtain the same level of downstream performance.

---

> ### Comment · Reviewer_xYFT · 2024-12-03
> **Response to authors**
>
> Thanks for your reply, and thanks for answering my questions during the rebuttal period. I’m raising my score (3–>5) with the expectation that, if the paper is accepted, then it will clearly state **in the abstract and the intro** that distilling into low rank is the method that performs on par or better.

---

> > ### Author Response · Authors · 2024-12-04
> > **Thanks for the reviewer**
> >
> > Thanks for the feedback and discussion, it helped to improve the paper. We have already added the requested change in the abstract and intro, it will be present in the next version of the manuscript.

---

### Official Review · Reviewer_PhpB · 2024-11-03

**Soundness:** 3
**Presentation:** 3
**Contribution:** 2
**Rating:** 6
**Confidence:** 2

**Summary:**

They present an approach for training low-rank autoregressive reward models for controlled generation. They first validate the low rank structure in standard but costly approaches to reward modeling -- which require taking a separate forward pass for each vocabulary item. They then propose a method which can distill this low rank structure into an autoregressive reward model which can operate much more efficiently. They finally, validate their method on toxicity and sentiment control tasks, using both reward and fluency (perplexity / MAUVE) as evaluation metrics.

**Strengths:**

* They do good analysis on the low rank structure of reward models
* They explain their method clearly and conduct a variety of experiments to validate it
* The paper is mostly well written
* Better controlled generation is an important problem.

**Weaknesses:**

* They don't clearly explain how their method is that different from others which predict rewards auto regressively. e.g. why is this approach different from others like Liu et al. and Krause et al. which effectively do the same thing? It seems that their approach outperforms these? Is this due to the specifics of their distillation objective? This could be more clearly spelled out and also ablated with experiments in the paper.

**Questions:**

* Is the W matrix in WE, meant to be lower rank than h?
* Why MSE loss and not binary cross entropy with soft labels?

---

> ### Author Response · Authors · 2024-11-20
> **Response**
>
> We thank the reviewer for their feedback. We are happy they like our analysis and that they find our method and experiments clear!
>
> > They don't clearly explain how their method is that different from others which predict rewards auto regressively. e.g. why is this approach different from others like Liu et al. and Krause et al. which effectively do the same thing?
>
> We agree that we should better highlight the difference between our method and work of Liu et al. (DExperts) and Krause et al. (GeDi). We look closer into parametrization of the output layer and propose a novel parametrization of the output layer to decouple the prefix score from marginal scores of next tokens (Eq. 6), instead of predicting everything in one go; and for this parametrization, we propose a regularization (Eq. 11), which makes it easier for the model to abstain.
>
> Additinally, DExperts and  GeDi are trained with the language modeling objective, while our method follows a reward modeling approach. As Deng et al. (RAD) show, the reward modeling approach leads to better fluency and control, which we also highlight in the Figure 9 in Appendix (see updated pdf).
>
> We clarify this in the updated version.
>
> > Is the W matrix in WE, meant to be lower rank than h?
>
> We don’t mean W to learn the low rank structure. In our setup, W is needed to support the multi-task objective of RAD (section 4.2), where we introduce a separate $W_i$ for each of the toxicity types.
>
> >Why MSE loss and not binary cross entropy with soft labels?
>
> We agree that cross-entropy is a more straightforward choice. We decided to use squared loss to closely follow the RAD approach. We add an additional ablation experiment using binary cross-entropy loss (Appendix: section F.2), where we observe that ARM trained with cross-entropy loss has slightly worse performance for the detoxification task. Additionally, having the squared loss allows us to analyze the training objective from a simple weighted average perspective (see Equation 5)
>
> We hope our response addresses your concerns.

---

> > ### Comment · Reviewer_PhpB · 2024-11-26
> >
> > Thank you for the response. I appreciate the clarifications you added regarding how your work differs from the prior works. I appreciate the new experiment with cross entropy loss as well. I will keep my score at a 6.

---

### Official Review · Reviewer_BihH · 2024-11-04

**Soundness:** 3
**Presentation:** 3
**Contribution:** 3
**Rating:** 8
**Confidence:** 3

**Summary:**

This work present a low-rank approximation of RAD for controlled decoding of LMs. The paper first analyzes RAD, a well-known constrained decoding method and finds that RAD requires model call for all possible tokens making it computationally slow. An SVD analysis of a N x |V| reward matrix indicates that the reward model learns a low rank approximation. Based on this, the paper proposes ARM that learns a low rank approximation and computes the reward for all possible output tokens at once resulting in better computational efficiency. The paper performs evaluations on both toxicity and sentiment control showing that this method performs similar to RAD while being significantly faster.

**Strengths:**

- This paper shows a good analysis that RAD learns a low-rank approximation and therefore similar performance on toxicity/sentiments can be obtained with low-rank approximation. The reward model is split into two parts -- one for baseline reward estimation for the provided prefix and a separate term for the next token, allowing for an efficient inference framework.
- The low rank approximation allows for the reward computation for all possible next tokens in a single forward pass. Therefore, ARM provides a computationally efficient way to compute the reward as shown by Figure 6.

**Weaknesses:**

- In terms of novelty, ARM essentially modifies the output head of the reward model to give the reward output for all K tokens at the same time for computational efficiency. This is a simple modification but the rank analysis presented in the paper make the claim to support this formulation stronger.
- The paper presents a linear approximation of the reward model for all K tokens simultaneously. It would be interesting to check how concatenation of $[Hw1^T; HWE]$ passed through an MLP (or any non-linear transformation) would perform. This should not increase the computational efficiency significantly but might improve the performance (especially to match RAD perplexity) for lower average maximal toxicity.

**Questions:**

- Could you comment on why the perplexity of ARM is higher in general for both the evaluations? Could adding an extra regularization term on natural text improve the perplexity?

**Details Of Ethics Concerns:**

Although this paper works with RealToxicityPrompts toxicity dataset, this is standard dataset and I do not believe an ethics review is required

---

> ### Author Response · Authors · 2024-11-20
> **Response**
>
> We thank the reviewer for their feedback. We are happy to hear that you find our analysis interesting!
> > Could you comment on why the perplexity of ARM is higher in general for both the evaluations? Could adding an extra regularization term on natural text improve the perplexity?
>
> We observe that ARM trained with distillation loss more closely matches the performance of RAD or even outperforms RAD for sentiment control task, while ARM trained on original responses is less fluent.
> One clear difference is that when training from data, we will see short contexts multiple times with different reward responses and must implicitly converge to their average, while in distillation, the teacher already performs this compression and provides a single deterministic target $\hat{r}(v|x)$ for every context $(x,v)$. We conjecture that this may lead to better-trained distilled models.
> We agree with you that it is possible that extra tuning or regularization would improve ARM trained on original responses. Thank you for the suggestion to try using MLP to non-linearly process the reward scores. We observe that MLP parametrization performs on par with the more simple linear parametrization (see Appendix F.2.2 for the ablation experiment).

---

### Author Response · Authors · 2024-11-20
**General response: summary of the revisions**

We thank all reviewers for their detailed feedback. We have uploaded a revised version of the manuscript, where we mark the changes in a different color. Below we summarize the updates:

- We moved Related work section higher in the text (Section 4), highlighted the connection to the reinforcement learning direction.
- Section 5.4. Added discussion of the difference between distilled ARM and ARM trained on original utterances.
--------
- Section 3.1. We highlighted our findings for the analysis of RAD and improved the formulations of our statements:
   1. RAD is capable to approximate $P_\Omega(R)$ matrix with high rank: $\text{rank}(P_\Omega(R)) > d$, where $d$ is the dimensionality of the model.
   2. We observe that the reward matrix learned by RAD tends to be low-rank.
   3. Data has low minimal rank.
- We clarify the Eq. 7. in Section 3.2.
- Section 3.3. We introduce the distillation task and clarify the impact of regularization.
- Figures 3,4. We simplify the figures and move the k=40 case to the Appendix. We clarify how to interpret the results in Section 5.4.
- We clarify the impact of regularization in Section 5.5.
- Appendix B.2. We restructure the section on data rank and provide more rigorous statements.
- Appendix C.4. We add discussion of potential issues with numerical rank estimation.
- Appendix F. We move the comparison of ARM and RAD with k=40 to the Appendix F from the main text to simplify the figures. We add additional ablation study results, as requested by the reviewers (cross-entropy loss vs squared loss, DExperts vs ARM comparison for the trade-off plot, and non-linear processing of the next token scores with MLP).

We are happy to address any further questions or concerns.

---

> ### Comment · Area_Chair_6SZZ · 2024-11-20
> **additional comments from AC**
>
> Dear authors,
>
> Thanks for the revisions and responses!
>
> I think the premise of improving efficiency in controlled generation is shared with many other works and the q-function view of the decoder (i.e., predicting the value of the next token for all alphabet in one forward pass) is shared between many other works that needs more substantiation beyond what is done currently:
>
> - Mudgal, S., Lee, J., Ganapathy, H., Li, Y., Wang, T., Huang, Y., Chen, Z., Cheng, H.T., Collins, M., Strohman, T. and Chen, J., 2023. Controlled decoding from language models. arXiv preprint arXiv:2310.17022.
>
> - Han, S., Shenfeld, I., Srivastava, A., Kim, Y. and Agrawal, P., 2024. Value Augmented Sampling for Language Model Alignment and Personalization. arXiv preprint arXiv:2405.06639.
>
> - Chakraborty, S., Ghosal, S.S., Yin, M., Manocha, D., Wang, M., Bedi, A.S. and Huang, F., 2024. Transfer Q Star: Principled Decoding for LLM Alignment. arXiv preprint arXiv:2405.20495.
>
> Also, there is work on parameter efficient learning of rewards and policies that seems related:
>
> - Sidahmed, H., Phatale, S., Hutcheson, A., Lin, Z., Chen, Z., Yu, Z., Jin, J., Komarytsia, R., Ahlheim, C., Zhu, Y. and Chaudhary, S., 2024. PERL: Parameter Efficient Reinforcement Learning from Human Feedback. arXiv preprint arXiv:2403.10704.
>
> Best,\
> AC

---

> > ### Author Response · Authors · 2024-11-22
> > **Response to AC and thank you for the valuable suggestions**
> >
> > We thank the AC for the valuable references. We indeed acknowledge the relationship with value function and q-function style parametrizations from this line of work; we emphasized this in the revision.
> >
> > We highlight that there is no consensus on which of the two parametrizations is best to use in text generation scenarios. Cao et al 2023 use the q-function parametrization, while Mudgal et al 2024, Chakraborty et al 2024 parametrize the value function without discussing the alternative or comparing to it.
> >
> > The most relevant discussion is the one in the preprint of Han et al 2024 **(Appendix B)**, where they compare the Value-function parametrized model to the Q-function parametrized model. Their analysis brings interesting connections to our work but **leads to different conclusions**: they find inferior performance of q-function parametrization compared to value-function parametrization. To highlight our contributions, we first analyze the full rank v-function style parametrization, which is assumed to be better because it's more expressive, and we actually investigate and show that yes it can capture high-rank solutions but it does not do so in practice.
> >
> > Additionally, we will include a reference to the Dueling Networks discussed in Han et al 2024, which is similar to our choice of the decomposed linear parametrization.
> >
> > - Chakraborty et al 2024. Transfer Q*: Principled Decoding for LLM Alignment. https://openreview.net/forum?id=5PrShrKxoX
> > - Han et al 2024, Value Augmented Sampling for Language Model Alignment and Personalization. https://arxiv.org/abs/2405.06639/
> > - Mudgal et al 2024. Controlled Decoding from Language Models. https://openreview.net/pdf?id=bVIcZb7Qa0
> > - Cao et al. 2023. Systematic Rectification of Language Models via Dead-end Analysis. https://arxiv.org/abs/2302.14003
> > - Tang et al 2024.  VA-learning as a more efficient alternative to Q-learning” https://arxiv.org/abs/2305.18161

---

### Meta-Review · Area_Chair_6SZZ · 2024-12-23

**Metareview:**

This work proposes autoregressive reward model (ARM) which is o distill reward augmented decoding (RAD) (Deng and Raffel, 2023) into a new model. The paper showcases effectiveness of the approach. As such, the proposal of this paper is a two-step process where first RAD needs to be trained and then distilled into a new model using soft distillation. The experiments show that ARM is effective at preserving the effectiveness of RAD and at times even surpasses it showcasing better generalization while it is more efficient at decoding time given that a single call to the q-network gives all logits that are needed at decoding time. The main selling point of ARM (which is the q-network decoding) has also appeared in (Mudgal et al., 2023) and (Han et al., 2024) as mentioned during the reviewer discussion-time by reviewers and the AC. In particular, Mudgal et al. (2023) train the q-function using the CD-Q method which relies on the Bellman operator instead of the online FUDGE-style (Yang and Klein, 2021) learning of the q-function, which is used here. The experiments of Mudgal et al. (2023) show the effectiveness of their approach in directly learning the q-function, hence it is crucial to compare the proposed method here with CD-Q approach of learning a q-function which offers the exact same inference-time efficiency, especially given that the proposal here requires a two-step process as opposed to the direct approach of (Mudgal et al., 2023) in order to better understand the performance of the distillation method proposed here. Finally, while not crucial for acceptance, it would be nice if the authors can remove the extra step of training RAD so that the entire process could be obtained in a single step. We hope the authors can take the comments of the reviewers and revise their paper for the next submission.

Deng, Haikang, and Colin Raffel. "Reward-augmented decoding: Efficient controlled text generation with a unidirectional reward model." arXiv preprint arXiv:2310.09520 (2023).

Mudgal, S., Lee, J., Ganapathy, H., Li, Y., Wang, T., Huang, Y., Chen, Z., Cheng, H.T., Collins, M., Strohman, T. and Chen, J., 2023. Controlled decoding from language models. arXiv preprint arXiv:2310.17022.

Yang, Kevin, and Dan Klein. "FUDGE: Controlled text generation with future discriminators." arXiv preprint arXiv:2104.05218 (2021).

Han, S., Shenfeld, I., Srivastava, A., Kim, Y. and Agrawal, P., 2024. Value Augmented Sampling for Language Model Alignment and Personalization. arXiv preprint arXiv:2405.06639.

**Additional Comments On Reviewer Discussion:**

The reviewers mentioned that the novelty of the paper is limited, which we agreed is not a blocker for publication. The reviewers and AC also agree that the distillation of RAD into a more efficient q-network is compelling. However, the paper still needs to be situated better with respect to the literature before it can be published.

---

### Decision · Program_Chairs · 2025-01-22

Reject